# Learning Adaptive Multi-Stage Energy-based Prior for Hierarchical Generative Model

**Jiali Cui**                                                                    *jcui7@stevens.edu*
*Department of Computer Science*
*Stevens Institute of Technology*

**Tian Han**                                                                     *than6@stevens.edu*
*Department of Computer Science*
*Stevens Institute of Technology*

**Reviewed on OpenReview:** *https://openreview.net/forum?id=W2zqUkA9Ub*

## Abstract

Hierarchical generative models represent data with multiple layers of latent variables organized in a top-down structure. These models typically assume Gaussian priors for multi-layer latent variables, which lack expressivity for the contextual dependencies among latents, resulting in a distribution gap between the prior and the learned posterior. Recent works have explored hierarchical energy-based prior models (EBMs) as a more expressive alternative to bridge this gap. However, most approaches learn only a *single* EBM, which can be ineffective when the target distribution is highly multi-modal and multi-scale across hierarchical layers of latent variables. In this work, we propose a framework that learns *multi-stage* hierarchical EBM priors, where a sequence of adaptive stages progressively refines the prior to match the posterior. Our method supports both joint training with the generator and a two-phase strategy for deeper hierarchies. Experiments across standard benchmarks show that our approach consistently generates higher-quality images and learns effective hierarchical representations.

## 1 Introduction

Hierarchical generative models, also known as multi-layer generator models, represent data using multiple layers of latent variables arranged in a top-down structure. Top layers typically capture high-level semantics, while bottom layers capture low-level fine details. Such models have shown strong performance in modelling complex data distributions and learning multi-level representations (Child, 2021; Vahdat & Kautz, 2020; Havtorn et al., 2021; Maaløe et al., 2019). The prior distributions over these latent variables are usually assumed to be Gaussian, which mainly capture inter-layer relationships (i.e., dependencies between layers) and ignore intra-layer contextual relationships (i.e., dependencies among latent units within the same layer). This lack of expressivity often leads to a mismatch between the assumed prior and the aggregated posterior learned from data, ultimately degrading the quality of generated samples and the capability of learning hierarchical representations.

Recent advances have introduced energy-based models (EBMs) as more expressive priors (Cui et al., 2023a;b; Aneja et al., 2021). EBMs can model rich intra-layer dependencies, offering a stronger inductive bias and modelling capacity than Gaussians. However, most existing approaches consider only a single EBM ("single-stage") for the complex posterior. When the posterior distribution is highly multi-modal and latent scales differently across layers, learning a single EBM prior is difficult and often ineffective. One line of research seeks to overcome this by borrowing ideas from diffusion models (Cui & Han, 2024; Yu et al., 2022; 2024), which introduce a sequence of conditional EBMs learned at different noise levels. While effective, these approaches depend on carefully designed diffusion schedules and require costly sampling procedures. Another

line of work (Xiao & Han, 2022; Rhodes et al., 2020) proposes multi-stage learning for marginal EBMs (Xiao & Han, 2022), which develops a self-adaptive strategy that evolves multiple EBMs without relying on a fixed schedule. Yet, these methods are restricted to *flat* latent spaces, limiting their ability to model hierarchical structures and capture layered representations. As a result, the development of multi-stage hierarchical EBM priors remains largely unexplored.

In this work, we develop multi-stage learning for hierarchical EBM priors. Our framework constructs a sequence of hierarchical EBM priors, viewed as density-ratio estimators, where each stage adaptively refines the prior by treating the previous stage as the new base distribution. This formulation constructs a chain of ratio bridges that progressively align the prior with the posterior, yielding a more robust and tractable solution than single-step methods. Unlike diffusion-based approaches, our method does not require a pre-defined schedule, but instead allows both the generator and hierarchical EBMs to evolve self-adaptively. Compared to flat latent space Adaptive-CE, the multi-scale nature of hierarchical latent space makes sampling across different layers difficult. To address this, we adopt a uni-scale latent space reparameterization (Section 3.2), which transforms the multi-scale latent variables into a consistent space where EBMs can be learned and sampled more effectively. Our framework supports two training schemes: (i) *joint learning*, where the generator and EBMs are optimized together, and (ii) *two-phase* strategy for deeper hierarchies, where the generator is trained first and the EBMs are learned afterward as complementary priors.

Extensive experiments across benchmark datasets show that our multi-stage hierarchical EBM prior not only improves image quality but also learns semantically meaningful, layered representations. Ablation studies further validate the benefits of our proposed multi-stage learning strategy. Together, these results demonstrate that multi-stage hierarchical EBMs provide a powerful and scalable alternative to Gaussian or single-step EBM priors in hierarchical generative modelling.

In summary, our main contributions are as follows:

- We propose a novel framework that introduces multi-stage EBM priors for hierarchical generative models, enabling more expressive latent modelling than Gaussian or single-stage EBM prior.

- We adopt a uni-scale latent space reparameterization and develop two training schemes, which improve the effectiveness and efficiency of EBM learning and sampling.

- We perform extensive experiments on standard benchmarks, demonstrating that our approach consistently generates higher-quality samples and learns effective hierarchical representations.

## 2 Preliminary

### 2.1 Hierarchical Generative Model

Hierarchical generative models extend the standard latent variable generative model by introducing multiple layers of latent variables arranged in a top-down structure. Each layer captures information at a different level of abstraction, with higher layers encoding global semantics and lower layers modelling fine details. Formally, let $\mathbf{X} \in \mathbb{R}^D$ denote an observed data example in a high-dimensional space, and let $\mathbf{z} \in \mathbb{R}^d$ denote latent variables with $d \ll D$. A hierarchical model defines a joint distribution over $\mathbf{X}$ and a hierarchy of latent variables $\mathbf{Z} = \{\mathbf{z}_1, \ldots, \mathbf{z}_L\}$. For example, this can be done through factorization Sønderby et al. (2016a); Nijkamp et al. (2020):

$$
\begin{aligned}
p_\beta(\mathbf{X}, \mathbf{Z}) =& p_{\beta_0}(\mathbf{X}|\mathbf{Z})p_{\beta_{>0}}(\mathbf{Z}) \quad \text{where} \\
p_{\beta_{>0}}(\mathbf{Z}) =& \prod_{i=1}^{L-1} p_{\beta_i}(\mathbf{z}_i|\mathbf{z}_{i+1})p_0(\mathbf{z}_L)
\end{aligned}
\tag{1}
$$

$p_{\beta_0}(\mathbf{X}|\mathbf{Z})$ is the generation model parameterized by $\beta_0$, while $p_{\beta_{>0}}(\mathbf{Z})$ specifies the hierarchical prior over latent variables. Each conditional dependency $p_{\beta_i}(\mathbf{z}_i|\mathbf{z}_{i+1})$ is modeled as a Gaussian distribution $\mathcal{N}(\mathbf{z}_i; \mu_{\beta_i}(\mathbf{z}_{i+1}), \sigma^2_{\beta_i}(\mathbf{z}_{i+1}))$, where $\mu_{\beta_i}$ and $\sigma^2_{\beta_i}$ denote the mean and diagonal covariance determined by $\mathbf{z}_{i+1}$. At the top of the hierarchy, the prior $p_0(\mathbf{z}_L)$ is typically set to a standard Gaussian.

Although hierarchical generative models provide a powerful framework for learning layered representations through multiple latent variables, their effectiveness is constrained by Gaussian priors. Such priors primarily capture dependencies across layers in the top-down hierarchy, but they fail to model the richer relationships among latent variables within the same layer, e.g., latent units at the same layer maybe distributed conditionally independent. This limited expressiveness leads to a mismatch between the assumed prior and the true aggregated posterior, commonly referred to as the prior-hole problem, resulting in degraded generation quality and learned representations.

## 2.2 Hierarchical Energy-based Prior

Energy-based models (EBMs) are well known for their ability to capture complex dependencies, especially within high-dimensional data or latent spaces (Xie et al., 2016; Nijkamp et al., 2019; Gao et al., 2021; Pang et al., 2020a; Du et al., 2021). When applied to hierarchical generative models, EBMs can act as expressive priors that model richer intra-layer interactions among latent variables, such as NCP-VAE Aneja et al. (2021) and JointEBM Cui et al. (2023a). In particular, JointEBM proposes a hierarchical EBM prior over all latent variables as

$$p_{\omega,\beta_{>0}}(\mathbf{Z}) = \frac{1}{\mathcal{Z}_{\omega,\beta_{>0}}} \exp\left[F_\omega(\mathbf{Z})\right] p_{\beta_{>0}}(\mathbf{Z}) \qquad (2)$$

where the energy function $F_\omega(\mathbf{Z}) = \sum_{i=1}^{L} f_\omega(\mathbf{z}_i)$ couples the latent variables across all layers, $p_{\beta_{>0}}(\mathbf{Z})$ is the Gaussian prior defined in Eqn 1, and $\mathcal{Z}_{\omega,\beta_{>0}}$ is the normalizing constant.

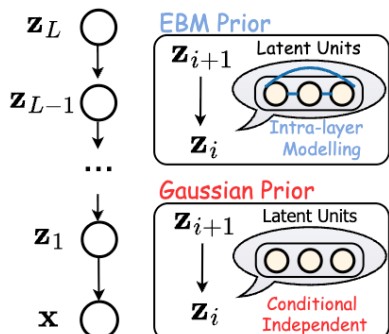

Figure 1: Multi-layer latent variables and latent units.

For learning such hierarchical EBM prior, the learning objective can be viewed as minimizing the KL divergence between the aggregated posterior $q(\mathbf{Z})$ and the EBM prior $p_{\omega,\beta_{>0}}(\mathbf{Z})$:

$$D_{\mathrm{KL}}(q(\mathbf{Z})||p_{\omega,\beta_{>0}}(\mathbf{Z})) \quad \text{where} \quad q(\mathbf{Z}) = \int p_\beta(\mathbf{Z}|\mathbf{X}) p_{\mathrm{data}}(\mathbf{X}) d\mathbf{X} \qquad (3)$$

Although this hierarchical EBM prior demonstrates the capability of modelling intra-layer contextual relationships (see Fig.1), providing greater expressivity and capacity than Gaussian priors, learning remains challenging. Specifically, this approach relies on a single-stage EBM prior to bridge the gap between the aggregated posterior $q(\mathbf{Z})$ and the base Gaussian prior $p_{\beta_{>0}}(\mathbf{Z})$. However, because $q(\mathbf{Z})$ is typically highly multi-modal, forcing a single EBM to approximate the entire distribution is difficult and often leads to suboptimal results.

## 3 Methodology

### 3.1 Multi-Stage Density Ratio Estimation

**Single-Stage Density Ratio Estimation.** Given a target distribution $q(\mathbf{Z})$ and a base distribution $p(\mathbf{Z})$, the goal is to estimate their density ratio $r_\omega(\mathbf{Z}) = q(\mathbf{Z})/p(\mathbf{Z})$[1] Gutmann & Hyvärinen (2010). A binary classifier $D_\omega(\mathbf{Z}) : \mathbb{R}^d \to (0,1)$ can be trained to discriminate between samples from $q(\mathbf{Z})$ and $p(\mathbf{Z})$ using the standard binary cross-entropy loss:

$$-\mathbb{E}_{q(\mathbf{Z})}[\log D_\omega(\mathbf{Z})] - \mathbb{E}_{p(\mathbf{Z})}[\log(1 - D_\omega(\mathbf{Z}))] \qquad (4)$$

At optimum, $D_\omega^*(\mathbf{Z}) = \frac{q(\mathbf{Z})}{q(\mathbf{Z}) + p(\mathbf{Z})}$, yielding the estimator $r_\omega(\mathbf{Z}) = q(\mathbf{Z})/p(\mathbf{Z}) \approx \frac{D_\omega^*(\mathbf{Z})}{1 - D_\omega^*(\mathbf{Z})}$. We can also maximize an equivalent objective $\mathcal{L}(\omega)$ with respect to $r_\omega(\mathbf{Z})$ as

$$\mathbb{E}_{q(\mathbf{Z})}\left[\log \frac{r_\omega(\mathbf{Z})}{1 + r_\omega(\mathbf{Z})}\right] + \mathbb{E}_{p(\mathbf{Z})}\left[\log \frac{1}{1 + r_\omega(\mathbf{Z})}\right] \qquad (5)$$

---

[1] We assume access to samples from both $q(\mathbf{Z})$ and $p(\mathbf{Z})$.

This can be viewed as learning an EBM of the form $p_\omega(\mathbf{Z}) = \frac{1}{\mathcal{Z}} r_\omega(\mathbf{Z}) p(\mathbf{Z})$, where $\mathcal{Z}$ is the normalizing constant and $r_\omega(\mathbf{Z})$ is an unconstrained positive function. This procedure is widely known as noise-contrastive estimation (NCE). However, when $q(\mathbf{Z})$ and $p(\mathbf{Z})$ differ substantially, single-stage NCE often fails to provide reliable density ratio estimates (Xiao & Han, 2022).

**Multi-Stage Density Ratio Estimation.** To overcome the limitations of single-stage estimation, the density ratio can be learned in multiple stages. A sequence of estimators $\{r_{\omega_s}(\mathbf{Z})\}_{s=1}^m$ is constructed, where each stage progressively refines the distribution obtained from the previous one. Formally, the model distribution after $m$ stages is defined as

$$p_{\omega_m}(\mathbf{Z}) = r_{\omega_m}(\mathbf{Z}) \cdot p_{\omega_{m-1}}(\mathbf{Z}) = \prod_{s=1}^m r_{\omega_s}(\mathbf{Z}) \cdot p(\mathbf{Z}) \tag{6}$$

which can be viewed as a product of intermediate ratio bridges

$$\frac{q(\mathbf{Z})}{p(\mathbf{Z})} = \frac{q(\mathbf{Z})}{p_{\omega_m}(\mathbf{Z})} \cdot \frac{p_{\omega_m}(\mathbf{Z})}{p_{\omega_{m-1}}(\mathbf{Z})} \cdots \frac{p_{\omega_1}(\mathbf{Z})}{p(\mathbf{Z})} \tag{7}$$

This decomposition allows for smoother and more tractable learning, especially in the presence of highly multi-modal targets. Prior works have explored this idea in different forms: Rhodes et al. (2020) applied a fixed-stage design in data space, but achieved only limited success on simple datasets (e.g., MNIST). Xiao & Han (2022) introduced an adaptive multi-stage scheme in flat latent spaces, but such methods remain limited in their capacity to capture structured, hierarchical representations.

### 3.2 Toward Hierarchical Energy-based Prior

Directly applying multi-stage NCE to hierarchical latent spaces is non-trivial because of their multi-scale structure and strong inter-layer dependencies. While prior work has studied multi-stage learning in flat spaces (Xiao & Han, 2022), extending it to hierarchical EBMs requires addressing these additional challenges. Inspired by the reparameterization strategy introduced in VAEBM (Xiao et al., 2021), we adopt their uni-scale transformation that maps the original hierarchical latent space $\mathbf{Z}$ into a uni-scale space $\mathbf{W}$. Unlike VAEBM, which applies this idea to single-stage data-space EBM, we leverage the uni-scale space to enable efficient multi-stage learning for hierarchical EBM priors.

Recall that the Gaussian prior $p_{\beta_{>0}}(\mathbf{Z})$ in Eqn 1 is factorized into consecutive conditional Gaussians, $p_{\beta_i}(\mathbf{z}_i|\mathbf{z}_{i+1}) \sim \mathcal{N}(\mu_{\beta_i}(\mathbf{z}_{i+1}), \sigma_{\beta_i}^2(\mathbf{z}_{i+1}))$. Through the standard reparameterization trick, each latent variable can be expressed as $\mathbf{z}_i = \mu_{\beta_i}(\mathbf{z}_{i+1}) + \sigma_{\beta_i}(\mathbf{z}_{i+1}) \cdot \mathbf{w}_i$ where $\mathbf{w}_i \sim \mathcal{N}(0, I_{d_i})$. This defines an invertible mapping $\mathcal{T}_{\beta_{>0}}$ such that $\mathbf{Z} = \mathcal{T}_{\beta_{>0}}(\mathbf{W})$ and $\mathbf{W} = \mathcal{T}_{\beta_{>0}}^{-1}(\mathbf{Z})$. In the $\mathbf{W}$-space, the hierarchical EBM prior can then be expressed as $p_{\omega,\beta_{>0}}(\mathbf{W}) = r_\omega(\mathcal{T}_{\beta_{>0}}(\mathbf{W})) p_0(\mathbf{W})$ (see derivation in App. B.3), where $p_0(\mathbf{W})$ is a standard Gaussian with independent components $p_0(\mathbf{w}_i) \sim \mathcal{N}(0, I_{d_i})$.

Building on this representation, we define our multi-stage hierarchical EBM prior as

$$\text{On } \mathbf{Z}\text{-Space: } p_{\omega_m,\beta_{>0}}(\mathbf{Z}) = \prod_{s=1}^m r_{\omega_s}(\mathbf{Z}) \cdot p_{\beta_{>0}}(\mathbf{Z}) \quad \text{On } \mathbf{W}\text{-Space: } p_{\omega_m,\beta_{>0}}(\mathbf{W}) = \prod_{s=1}^m r_{\omega_s}(\mathcal{T}_{\beta_{>0}}(\mathbf{W})) \cdot p_0(\mathbf{W}) \tag{8}$$

On $\mathbf{Z}$-Space, the base distribution is Gaussian prior, and the multi-stage modelling aims to approximate the target distribution adaptively and progressively. On $\mathbf{W}$-Space, this formulation leverages the uni-scale reparameterization to make multi-stage learning feasible in hierarchical EBMs. The transformation $\mathcal{T}$ preserves hierarchical dependencies, while the $\mathbf{W}$-space provides a consistent representation that enables better training and sampling. We refer to App.B.3 for more details.

### 3.3 Learning Multi-Stage Hierarchical Energy-based Prior

#### 3.3.1 Joint Learning

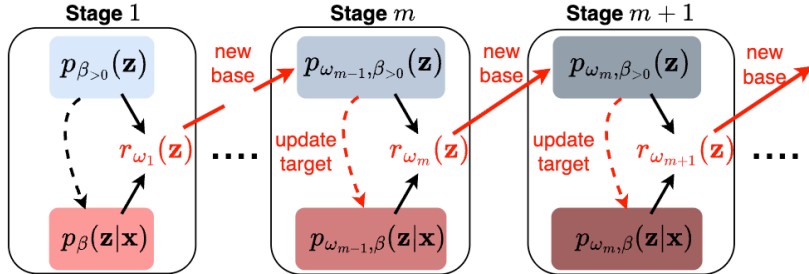

Figure 2: Updating schema of **joint learning**. At each stage $m$, a density ratio $r_{\omega_m}(\mathbf{Z})$ is estimated by contrastive learning, distinguishing samples from the prior and from the aggregated posterior. Posterior samples are obtained via MCMC posterior sampling with the prior of the current stage (dashed line), while prior samples can be taken from posterior samples at the previous stage. The estimated ratio is then used to update the prior, which becomes the base distribution for the next stage. By repeating this process, the prior is progressively refined across stages, enabling stable joint learning of both the generator and the hierarchical EBM prior.

With hierarchical EBM prior, we can specify a joint distribution of the hierarchical generative model as

$$p_{\omega_m,\beta}(\mathbf{X},\mathbf{Z}) = p_{\beta_0}(\mathbf{X}|\mathbf{Z})p_{\omega_m,\beta_{>0}}(\mathbf{Z}) \tag{9}$$

which involves generator parameters $\beta$ and multiple stages of EBM parameters $\omega$. Learning can be achieved by iterating between the maximum likelihood estimation (MLE) of $\beta$ and contrastive estimation of $\omega$.

Specifically, at the $m$-th stage, updating the generator parameters $\beta$ corresponds to maximizing the data likelihood. The gradient takes the form

$$\mathbb{E}_{p_{\text{data}}(\mathbf{X})p_{\omega_m,\beta}(\mathbf{Z}|\mathbf{X})}\left[\frac{\partial}{\partial\beta}\log p_{\beta_0}(\mathbf{X}|\mathbf{Z})\right] + \mathbb{E}_{p_{\text{data}}(\mathbf{X})p_{\omega_m,\beta}(\mathbf{Z}|\mathbf{X})}\left[\frac{\partial}{\partial\beta}\log p_{\beta_{>0}}(\mathbf{Z})\right] \tag{10}$$

where $p_{\omega_m,\beta}(\mathbf{Z}|\mathbf{X})$ denotes the generator posterior. These expectations are approximated by posterior sampling using Markov Chain Monte Carlo (MCMC), such as Langevin dynamics (see Eqn 14).

The EBM parameters $\omega_m$ are updated via contrastive estimation, where the generator posterior serves as the target distribution and the prior from the previous stage serves as the base distribution:

$$\mathbb{E}_{p_{\text{data}}(\mathbf{X})p_{\omega_{m-1},\beta}(\mathbf{Z}|\mathbf{X})}\left[\frac{\partial}{\partial\omega_m}\log\frac{r_{\omega_m}(\mathbf{Z})}{1+r_{\omega_m}(\mathbf{Z})}\right] + \mathbb{E}_{p_{\omega_{m-1},\beta}(\mathbf{Z})}\left[\frac{\partial}{\partial\omega_m}\log\frac{1}{1+r_{\omega_m}(\mathbf{Z})}\right] \tag{11}$$

where samples from $p_{\omega_{m-1},\beta}(\mathbf{Z})$ are also obtained using Langevin dynamics.

Since the hierarchical latent variables $\mathbf{Z}$ are inherently multi-scale, both posterior sampling and EBM sampling are carried out in the uni-scale $\mathbf{W}$-space. The results are then mapped back to $\mathbf{Z}$-space through the transformation function $\mathcal{T}$ for gradient computation. This joint learning procedure alternates between updating $\beta$ and $\omega$ at each stage, gradually refining both the generator and the hierarchical EBM prior. More efficiently, once the density ratio estimator achieves sufficient accuracy, the prior samples from $p_{\omega_{m-1},\beta}(\mathbf{Z})$ can be replaced by samples from the previous posterior $p_{\omega_{m-2},\beta}(\mathbf{Z}|\mathbf{X})$, thereby bypassing MCMC sampling from the prior and reducing computational cost (Xiao & Han, 2022).

While joint learning provides a principled way to train both the generator and the hierarchical EBM prior together, it relies heavily on MCMC posterior sampling. This procedure becomes increasingly inefficient as the hierarchy deepens, since each MCMC step requires backpropagation through all latent layers of the generator.

### 3.3.2 Two-phase Learning for Deep Hierarchy

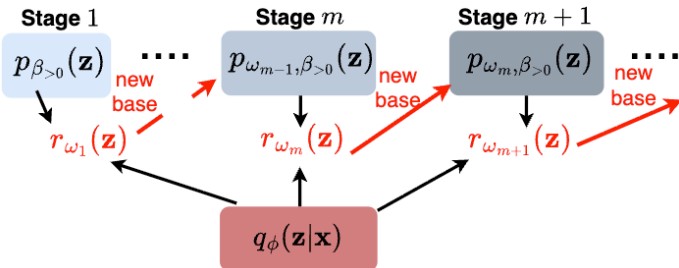

Figure 3: Updating schema of **two-phase** learning. In the second phase, the variational posterior $q_\phi(\mathbf{Z}|\mathbf{X})$ serves as the *fixed* target distribution for hierarchical EBM prior learning. At each stage $m$, a density ratio $r_{\omega_m}(\mathbf{Z})$ is estimated between $q_\phi(\mathbf{Z}|\mathbf{X})$ and the current base prior. The updated prior $p_{\omega_m,\beta_{>0}}(\mathbf{Z})$ is then used as the base distribution for the next stage.

To address this computational bottleneck, we introduce a two-phase learning strategy. In the first phase, we approximate the generator posterior using a variational inference model, which eliminates the need for costly posterior MCMC. In the second phase, our multi-stage EBM prior is learned on top of this fixed variational posterior, retaining the expressiveness of the EBM while ensuring scalability to deep hierarchical structures.

**First-phase: Variational Learning with Gaussian Prior.** Variational learning Sønderby et al. (2016a) introduces an inference model $q_\phi(\mathbf{Z}|\mathbf{X})$ that approximates the generator posterior distribution. For hierarchical latent variables, the inference model is also structured hierarchically, i.e., $q_\phi(\mathbf{Z}|\mathbf{X}) = q_{\phi_1}(\mathbf{z}_1|\mathbf{X}) \prod_{i=2}^{L} q_{\phi_i}(\mathbf{z}_i|\mathbf{z}_{i-1})$, where each conditional factor is modeled as a Gaussian distribution.

With such inference model, variational learning proceeds by maximizing the evidence lower bound (ELBO)

$$\mathbb{E}_{p_{\text{data}}(\mathbf{X})q_\phi(\mathbf{z}|\mathbf{X})} \left[ \log \frac{p_\beta(\mathbf{X},\mathbf{Z})}{q_\phi(\mathbf{Z}|\mathbf{X})} \right] = \mathbb{E}_{p_{\text{data}}(\mathbf{X})q_\phi(\mathbf{z}|\mathbf{X})} \left[ \log p_{\beta_0}(\mathbf{X}|\mathbf{Z}) \right] - \mathbb{E}_{p_{\text{data}}(\mathbf{X})}[D_{\text{KL}}(q_\phi(\mathbf{Z}|\mathbf{X})||p_{\beta_{>0}}(\mathbf{Z}))] \quad (12)$$

By maximizing the ELBO, we effectively maximize a lower bound of the true log-likelihood. The gap between the ELBO and the likelihood is given by the KL divergence term, ELBO $= \log p_\beta(\mathbf{X}) - D_{\text{KL}}(q_\phi(\mathbf{Z}|\mathbf{X})||p_\beta(\mathbf{Z}|\mathbf{X}))$, which encourages the variational posterior to approximate the generator posterior. In other words, variational learning can be interpreted as approximate MLE, where the inference model provides a tractable surrogate for posterior sampling. Once the inference model is well trained, its variational posterior $q_\phi(\mathbf{Z}|\mathbf{X})$ serves as a reliable approximation of $p_\beta(\mathbf{Z}|\mathbf{X})$, and can therefore be adopted as the target distribution in the second phase of our hierarchical EBM prior learning.

**Second-phase: Contrastive Estimation with Fixed Generator.** In the second phase, we fix both the generator parameters $\beta$ and the inference network $\phi$, and focus on learning the hierarchical EBM prior. The training objective at stage $m$ is

$$\mathbb{E}_{p_{\text{data}}(\mathbf{X})q_\phi(\mathbf{z}|\mathbf{X})} \left[ \frac{\partial}{\partial \omega_m} \log \frac{r_{\omega_m}(\mathbf{Z})}{1+r_{\omega_m}(\mathbf{Z})} \right] + \mathbb{E}_{p_{\omega_{m-1},\beta}(\mathbf{z})} \left[ \frac{\partial}{\partial \omega_m} \log \frac{1}{1+r_{\omega_m}(\mathbf{Z})} \right] \quad (13)$$

where posterior samples are efficiently obtained from the fixed inference model. Conceptually, this phase isolates the task of refining the prior: it learns to bridge the gap between the aggregated posterior $q_\phi(\mathbf{Z})$ and the base Gaussian prior $p_{\beta_{>0}}(\mathbf{Z})$, without requiring costly posterior MCMC updates.

This two-phase design offers two key advantages. First, it makes training more effective in deep hierarchical settings by decoupling generator learning from EBM refinement. Second, unlike most existing methods (Aneja et al., 2021; Xiao et al., 2021; Cui et al., 2023a) that attempt to close this gap with a single-stage EBM, we decompose the process into multiple adaptive stages. Each stage provides refinement of the prior, yielding smoother convergence and a more faithful approximation of complex, multi-modal posteriors.

### 3.3.3 Markov Chain Monte Carlo Sampling

Both training strategies require access to hierarchical latent variables from either the generator posterior or the EBM prior. To do so, we adopt MCMC sampling, such as Langevin dynamics (Neal et al., 2011), in the uni-scale space. Specifically, for an arbitrary distribution $p(\mathbf{z})$, it is performed as

$$\mathbf{z}_{\tau+1} = \mathbf{z}_\tau + s \frac{\partial}{\partial \mathbf{z}_\tau} \log p(\mathbf{z}_\tau) + \sqrt{2s} U_\tau \tag{14}$$

where $\tau$ indexes the time step, $s$ is the step size and $U_\tau$ is Gaussian noise.

**MCMC for EBM Prior.** To sample from the hierarchical EBM prior, we apply Langevin dynamics in the uni-scale $\mathbf{W}$-space. The gradient term is computed as $\frac{\partial}{\partial \mathbf{W}_\tau} \left[ \sum_{s=1}^m \log r_{\omega_s}(\mathcal{T}_{\beta>0}(\mathbf{W}_\tau)) - \frac{\|\mathbf{W}_\tau\|^2}{2} \right]$, which combines contributions from all ratio functions $r_{\omega_s}$ together with the standard Gaussian reference prior.

**MCMC for Posterior.** For generator posterior, we also perform Langevin dynamics in uni-scale $\mathbf{W}$-space. The gradient term is computed as $\frac{\partial}{\partial \mathbf{W}_\tau} \left[ \log p_\beta(\mathbf{X}|\mathcal{T}_{\beta>0}(\mathbf{W}_\tau)) + \log p_{\omega_m,\beta>0}(\mathbf{W}_\tau) \right]$ (see derivation in App.B.3).

**Variational and MCMC Sampling.** MCMC-based posterior sampling requires backpropagation through the generation model, which is often large. When the hierarchy is shallow (e.g., 2 layers), the generator and the EBM prior can be trained jointly with MCMC posterior sampling. In this regime, the cost of gradient propagation remains manageable, and MCMC sampling provides more accurate posterior estimates and better learning signals for the latent variables. However, when the hierarchy becomes deep (e.g., 30 layers), MCMC posterior sampling becomes expensive. Joint optimization of both the hierarchical generator and the hierarchical EBM prior under these conditions is often unstable. Following prior work Xiao et al. (2021), we adopt a two-phase learning strategy in which a hierarchical variational inference model replaces the MCMC posterior sampler, leading to more tractable and stable training in deep hierarchical settings MCMC sampling in the uni-scale ensures effective sampling for both prior and posterior distributions, while the transformation $\mathcal{T}$ maps the results back to the original hierarchical latent space.

## 4 Related Work

**Hierarchical Generative Models.** Hierarchical generative models employ multiple layers of latent variables organized in a top-down structure and are typically parameterized with Gaussian priors (Vahdat & Kautz, 2020; Child, 2021; Sønderby et al., 2016a; Maaløe et al., 2019; Nijkamp et al., 2020). While convenient, Gaussian priors lack statistical expressivity, often leading to the prior-hole problem and degraded generation quality (Aneja et al., 2021; Cui et al., 2023a). To overcome this limitation, recent studies have explored expressive priors such as EBMs, but most approaches rely on single-stage learning. Our work extends this line of research by developing multi-stage hierarchical EBM priors that effectively capture complex posterior distributions while remaining scalable to deep hierarchies.

**Energy-based Models.** EBMs have been widely applied in data space for modeling images and other high-dimensional signals (Xie et al., 2016; Nijkamp et al., 2019; Du et al., 2021; Gao et al., 2021; Cui & Han, 2023; Xiao et al., 2021). Extensions to latent space exist in both flat (Pang et al., 2020a; Yu et al., 2024; Schröder et al., 2023; Yuan et al., 2024) and hierarchical settings (Aneja et al., 2021; Cui et al., 2023a;b), but these methods adopt single-stage estimation, limiting their ability to approximate multi-modal posteriors. Multi-stage training has recently been studied in flat latent spaces (Xiao & Han, 2022), showing that a sequence of density ratio estimators can improve learning stability. However, extending this idea to hierarchical latent spaces introduces new challenges due to scale differences and inter-layer dependencies. Our work addresses these challenges with a uni-scale reparameterization and develops a self-adaptive multi-stage framework for hierarchical EBMs, enabling robust and scalable training across deep latent structures.

**Diffusion Energy-based Models.** Both diffusion frameworks and multi-stage learning approximate complex distributions by constructing a sequence of intermediate models. Diffusion methods learn conditional EBMs (e.g., $p_\omega(\mathbf{W}_t|\mathbf{W}_{t+1})$) guided by a pre-defined forward noise process (Gao et al., 2021; Cui & Han, 2024). While effective, this conditioning restricts each EBM to local refinements and requires carefully designed schedules. In contrast, our multi-stage framework builds marginal models (e.g.,

$p_{\omega_m}(\mathbf{W}) = \prod_{s=1}^m r_{\omega_s}(\mathbf{W}) p_0(\mathbf{W}))$ in a self-adaptive manner, allowing each stage to refine the prior distribution without a fixed noise schedule. From a learning perspective, diffusion-based EBMs are typically trained with MLE, which requires costly inner-loop MCMC sampling at every step. Our approach leverages contrastive estimation, which optimizes density ratios and avoids repeated inner-loop MCMC.

# 5 Experiments

## 5.1 Latent Space Modeling

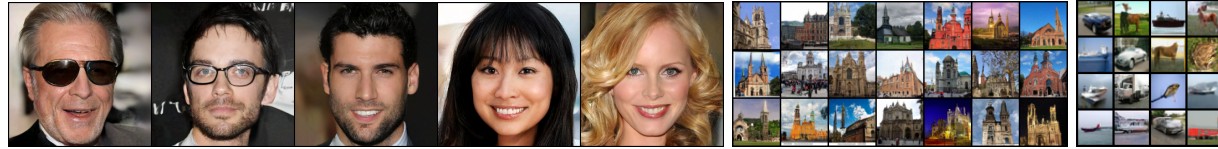

Figure 4: Synthesis on CelebA-HQ-256 (left), LSUN-Church-64 (center), and CIFAR10 (right).

We first evaluate our method on the task of latent space modeling. A well-learned hierarchical EBM prior should produce informative latent samples that, when passed through the generator, yield high-quality and realistic synthesis. Concretely, we generate images by sampling $\mathbf{W}$ from the learned EBM prior, mapping it into hierarchical latent variables $\mathbf{Z} = \mathcal{T}_{\beta>0}(\mathbf{W})$, and decoding with the generator $p_{\beta_0}(\mathbf{X}|\mathbf{Z})$. We benchmark across several datasets, including SVHN Netzer et al. (2011), CIFAR-10 Krizhevsky et al. (2009), CelebA-64 Liu et al. (2014), LSUN-Church-64 Yu et al. (2015), and high-resolution CelebA-HQ-256 Karras et al. (2017). Image quality is measured with Fréchet Inception Distance (FID) and Inception Score (IS).

Table 1: Joint learning for FID ($\downarrow$) comparison.

| Method | SVHN | CelebA-64 | CIFAR-10 |
|---|---|---|---|
| **Ours** | **20.42** | **29.12** | **61.12** |
| Adaptive CE Xiao & Han (2022) | 26.19 | 35.38 | 65.01 |
| Joint-EBM Cui et al. (2023a) | 26.81 | 33.60 | 66.32 |
| NCP-VAE Aneja et al. (2021) | 33.23 | 42.07 | 78.06 |
| LEBM Pang et al. (2020a) | 29.44 | 37.87 | 70.15 |
| ED-LEBM Schröder et al. (2023) | 28.10 | 36.73 | 73.58 |
| EBIPLA Marks et al. (2025) | 27.59 | 34.71 | 79.64 |
| EVaLP Dutta et al. (2025) | - | 35.90 | 76.43 |
| Diffusion-Amortized Yu et al. (2024) | 21.17 | 35.67 | 60.89 |
| 2s-VAE Dai & Wipf (2019) | 33.23 | 42.07 | 78.06 |
| BIVA Maaløe et al. (2019) | 31.65 | 33.58 | 66.37 |

Table 2: Two-phase learning for FID ($\downarrow$) comparison.

| Method | CelebA-HQ-256 | LSUN-Church-64 |
|---|---|---|
| **Ours** | **7.84** | **6.65** |
| Hierarchical Diff-EBM Cui & Han (2024) | 8.78 | 7.34 |
| Joint-EBM Cui et al. (2023a) | 9.89 | 8.38 |
| NCP-VAE Aneja et al. (2021) | 24.79 | - |
| NVAE* Vahdat & Kautz (2020) | 30.25 | 38.13 |
| NVAE*-Recon (rFID) | 1.64 | 2.45 |
| DRL-EBM ($T = 6$) Gao et al. (2021) | - | 7.04 |
| Adv-EBM Yin et al. (2020) | 17.31 | 10.84 |
| GLOW Kingma & Dhariwal (2018) | 68.93 | 59.35 |
| PGGAN Karras et al. (2017) | 8.03 | 6.42 |

Table 3: Two-phase learning on CIFAR-10.

| Methods | IS ($\uparrow$) | FID ($\downarrow$) |
|---|---|---|
| **Ours** | **9.03** | **7.80** |
| Hierarchical Diff-EBM Cui & Han (2024) | 9.03 | 8.93 |
| Joint-EBM Cui et al. (2023a) | 8.99 | 11.34 |
| NCP-VAE Aneja et al. (2021) | - | 24.08 |
| **Multi-layer Generator** | | |
| NVAE* Vahdat & Kautz (2020) | 5.30 | 37.73 |
| NVAE*-Recon (rFID) | - | 0.68 |
| HVAE Sønderby et al. (2016a) | - | 81.44 |
| BIVA Maaløe et al. (2019) | - | 66.37 |
| **Energy-based Models** | | |
| Dual-MCMC Cui & Han (2023) | 8.55 | 9.26 |
| Architectural-EBM Cui et al. (2023b) | - | 63.42 |
| DRL-EBM ($T = 6$) Gao et al. (2021) | 8.40 | 9.58 |
| Adaptive-CE Xiao & Han (2022) | - | 65.01 |
| VAEBM Xiao et al. (2021) | 8.43 | 12.19 |
| Hat EBM Hill et al. (2022) | - | 19.15 |
| ImprovedCD Du et al. (2021) | 7.85 | 25.1 |
| Divergence Triangle Han et al. (2020) | - | 30.10 |
| Adv-EBM Yin et al. (2020) | 9.10 | 13.21 |
| EVaLP Dutta et al. (2025) | - | 42.30 |
| **GANs+Score+Diffusion Models** | | |
| StyleGANv2 w/o ADA Karras et al. (2020) | 8.99 | 9.9 |
| Diffusion-Amortized Yu et al. (2024) | - | 57.72 |
| NCSN Song & Ermon (2019) | 8.87 | 25.32 |
| LSGM Vahdat et al. (2021) | - | 2.10 |
| DDPM ($T = 1000$) Ho et al. (2020) | 9.46 | 3.17 |

**Joint Learning.** We evaluate our framework under a joint training scheme with a two-layer ($L = 2$) hierarchical latent structure, where the generator and the multi-stage hierarchical EBM prior are learned jointly. We compare against several groups of baselines. (i) *Flat latent models with multi-stage EBMs*, such as Adaptive-CE Xiao & Han (2022), which learns multi-stage priors but only in single-layer latent spaces; (ii) *Hierarchical models with single-stage EBMs*, including Joint-EBM Cui et al. (2023a) and NCP-VAE Aneja et al. (2021), which extend EBMs to hierarchical structures but rely on a single-stage formulation;

(iii) *Other expressive priors or Gaussian-based hierarchical models*, such as LEBM Pang et al. (2020a), ED-LEBM Schröder et al. (2023), EBIPLA Marks et al. (2025), EVaLP Dutta et al. (2025), Diffusion-Amortized Yu et al. (2024), 2s-VAE Dai & Wipf (2019), and BIVA Maaløe et al. (2019).

As shown in Tab.1, our method consistently achieves superior results across datasets. These results suggest that introducing a multi-stage EBM prior into hierarchical latent spaces offers advantages over both single-stage EBMs and flat multi-stage approaches, resulting in stronger priors and improved generation quality.

**Two-Phase Learning.** For deep hierarchical structures, our multi-stage hierarchical EBM prior is learned as ratio bridges between the fixed aggregate posterior and the base Gaussian prior. Once learned, this EBM prior serves as a more informative alternative to the Gaussian prior, enabling generation of realistic samples.

Following the setup of prior works, including Hierarchical Diff-EBM Cui & Han (2024), Joint-EBM Cui et al. (2023a), and NCP-VAE Aneja et al. (2021), we build on top of NVAE Vahdat & Kautz (2020) as our backbone model (denoted NVAE$^*$). As shown in Tab.2 and Tab.3, our method achieves competitive or superior performance, even when compared to recent diffusion-based approaches. Notably, it yields substantial improvements over the NVAE$^*$ baseline, highlighting the efficacy of our multi-stage learning strategy in capturing complex latent distributions.

## 5.2 Analysis of Hierarchical Representation

A hallmark of hierarchical generative models is that different latent layers capture different levels of abstraction: bottom-layer variables tend to encode low-level visual attributes, while top-layer variables capture high-level semantic concepts. In this section, we demonstrate the capability of our multi-stage hierarchical EBM prior in learning hierarchical representations.

### 5.2.1 Hierarchical Controllable Synthesis

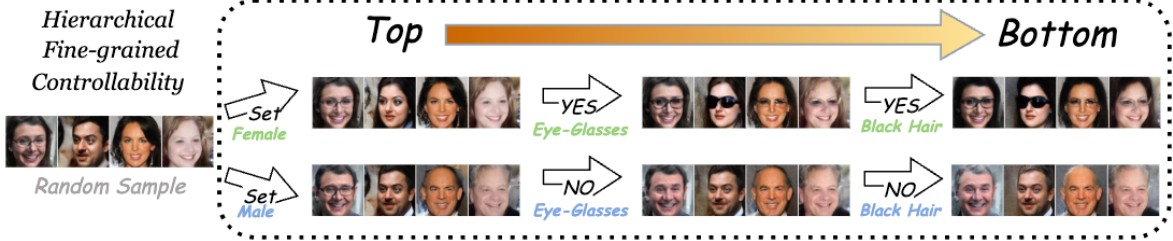

Figure 5: Starting from random samples, attributes such as gender, eyeglasses, and hair color can be manipulated in a top-down order. Higher-layer latents govern semantic structure (e.g., gender, identity), while lower-layer latents refine local appearance (e.g., eyeglasses, hair). Importantly, sampling at lower layers preserves the semantics imposed by higher layers, enabling interpretable and fine-grained controllability across the hierarchy.

An advantage of our framework is the controllability from the multi-stage EBM prior, even when the backbone generator is trained in a fully unsupervised manner. Inspired by (Salimans et al., 2016), which demonstrated how GAN discriminators can be repurposed for classification, we reinterpret each stage of our multi-stage NCE framework as a learned discriminator.

Given $K$ class labels or attributes (denoted by $y$), the hierarchical EBM prior can be extended into a multi-class classifier with $K + 1$ outputs, where the $(K + 1)$-th logit continues to distinguish posterior samples from prior samples. This formulation enables modeling of the joint distribution $p_{\omega_m, \beta_{\geq 0}}(\mathbf{W}, y)$ (see details in App.B.4), allowing conditional generation under a specified label $y$. As illustrated in Fig.5, our method leverages the discriminative structure of the multi-stage EBM prior to achieve fine-grained and semantically consistent controllability. The hierarchical organization further ensures that semantic structure imposed by higher layers is preserved when lower layers are sampled, enabling controllable generation that is both structured and coherent across different levels of abstraction.

### 5.2.2  Hierarchical Out-of-Distribution Detection.

Beyond generation and representation learning, our multi-stage hierarchical EBM prior also provides a natural mechanism for out-of-distribution (OOD) detection. Each stage of the prior operates as a density-ratio classifier, distinguishing samples from the aggregated posterior versus the base prior in latent space. As a result, OOD detection emerges as a built-in capability of our framework, requiring no additional supervision or auxiliary objectives.

We evaluate this property by training on CIFAR-10 as the in-distribution dataset and testing on SVHN as OOD. Detection is based on the classifier output $D : \mathbb{R}^d \to (0,1)$, where lower values indicate higher likelihood of being OOD. As shown in Table.4 and Figure.6, AUROC scores increase with depth: top-layer latents provide the strongest OOD signals by encoding distribution-specific semantics, while lower layers capture more generic features that are often shared across datasets Havtorn et al. (2021).

| AUROC | $L^{>0}$ | $L^{>3}$ | $L^{>6}$ | $L^{>9}$ | $L^{>12}$ |
|---|---|---|---|---|---|
| **Stage 1** | 0.3312 | 0.3858 | 0.4118 | 0.4312 | 0.4410 |
| **Stage 2** | 0.3501 | 0.3953 | 0.4315 | 0.4626 | 0.5235 |
| **Stage 3** | 0.4045 | 0.4775 | 0.4969 | 0.5351 | 0.5713 |
| AUROC | $L^{>15}$ | $L^{>18}$ | $L^{>21}$ | $L^{>24}$ | $L^{>27}$ |
| **Stage 1** | 0.5382 | 0.6683 | 0.6735 | 0.6960 | 0.7079 |
| **Stage 2** | 0.6416 | 0.7277 | 0.7335 | 0.7471 | 0.8002 |
| **Stage 3** | 0.6652 | 0.7835 | 0.7710 | 0.8469 | 0.8897 |

Table 4: AUROC (↑) for OOD detection. $L > k$ denotes for using the top $k$ layers. The total number of layers is 30.

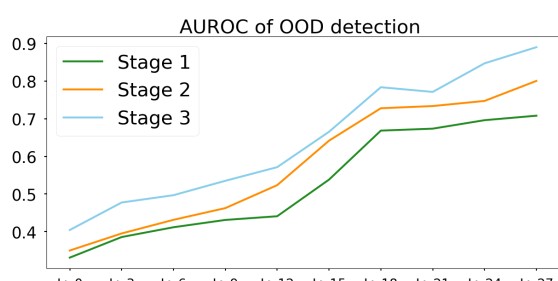

Figure 6: Visualization of AUROC curve.

These findings highlight a key advantage of our hierarchical design. By aligning representation depth with semantic specificity, our model not only improves generative modeling but also enhances robustness in detecting OOD samples.

## 5.3  Analysis of Multi-Stage Learning Dynamic

### 5.3.1  Multi-Stage Learning Analysis

We next analyze the learning behavior of our multi-stage framework. In our formulation, each stage of density ratio estimation refines the prior by building on the ratio estimator learned in the previous stage, effectively evolving the base distribution toward the target posterior. As the base distribution becomes progressively better aligned with the posterior, the discrimination task between posterior and prior samples naturally becomes harder. This increasing difficulty is reflected in higher loss values observed in later stages.

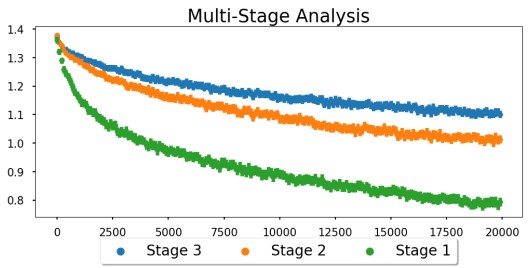

Figure 7: Visualization of stage-wise evolution with loss trends averaged across all latent layers.

Fig.7 illustrates this trend: the average loss (aggregated across all latent layers) grows as the stage index increases. This progression confirms that successive stages refine increasingly aligned priors, making the classification task progressively more challenging. In other words, earlier stages correct large mismatches between the prior and posterior, while later stages focus on finer adjustments. Such behavior is consistent with theoretical expectations and aligns with prior observations in flat latent settings Xiao & Han (2022). These results provide strong empirical support for the effectiveness of our multi-stage learning scheme.

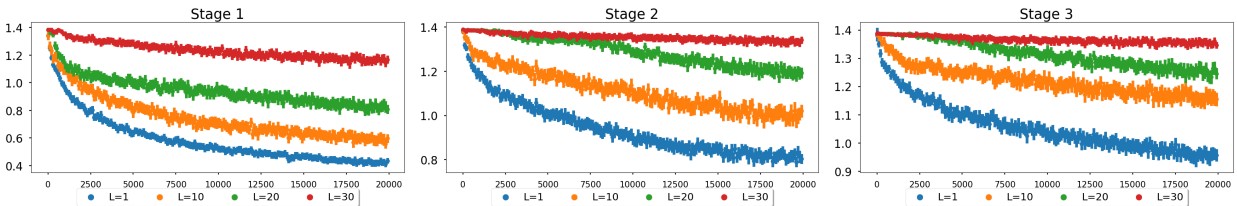

Figure 8: Visualization of layer-wise evolution from stage 1 to stage 3.

### 5.3.2 Multi-Layer Learning Analysis

To further examine learning dynamics, we analyze loss trends across different latent layers. In hierarchical latent structures, lower layers typically encode fine-grained, low-level details such as texture or background. These features are often more variable and complex, creating a larger gap between the aggregate posterior and the base prior. Consequently, the density ratio estimation task at lower layers is relatively easier, yielding lower loss values.

This trend is evident in Fig.8, where lower layers consistently report smaller losses than top layers. As additional learning stages are introduced, the multi-stage EBM prior progressively narrows the gap between the posterior and the base distribution at each layer. This refinement makes the classification task more difficult across all layers, leading to higher losses over time. The observed progression confirms that our multi-stage framework effectively reduces discrepancies at every level of the hierarchy, supporting consistent prior learning in complex multi-layer latent spaces.

### 5.4 Analysis of Energy Landscape

#### 5.4.1 Joint Langevin Transition.

We analyze the sampling behavior using Langevin transition. Starting from a sample drawn from the standard Gaussian, we perform 10 Langevin steps for learned multi-stage hierarchical EBM priors. As illustrated in Fig.9, image quality improves gradually and coherently with each step. This smooth evolution reflects the well-structured energy landscape of our model, which is further facilitated by operating in the uni-scale $\mathbf{w}$-space.

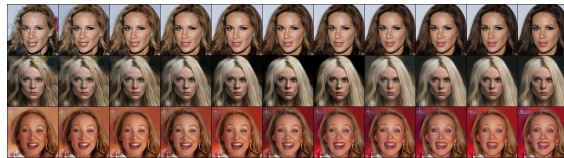

Figure 9: Joint Langevin transition.

#### 5.4.2 Progressive Transition

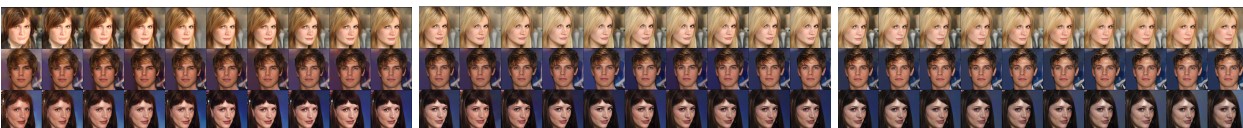

Figure 10: Visualization of progressive Langevin transition.

We next examine whether each stage of the EBM prior can serve as a base distribution for the subsequent stage. To this end, we perform progressive sampling: at the $s$-th stage, 10-step Langevin dynamics is applied only with the $s$-th stage EBM prior, initialized from the final sample of the $s-1$ stage. Fig.10 shows that each stage produces a refined update of the sample, demonstrating that the sequence of EBMs provides a coherent progression that incrementally shapes the energy landscape.

### 5.5 Ablation Studies

**Multi-Stage vs. Single-Stage under Fixed Iterations.** We begin by testing the effectiveness of the proposed multi-stage framework. To ensure fairness, we keep all other settings and fix the total training budget to 15K iterations. We then compare two settings: (i) a *single-stage* hierarchical EBM prior trained for the full 15K iterations, and (ii) a *three-stage* model, where 5K iterations are allocated to each stage.

As reported in Tab.5, the multi-stage approach substantially outperforms the single-stage baseline, achieving an FID of 9.48 compared to 15.15. Notably, the table also reveals a clear progression: Stage 1 starts weaker due to limited training, but each subsequent stage produces marked improvements, with the final stage delivering the strongest results. This experiment highlights our framework: rather than relying on prolonged training of a single-stage EBM prior, decomposing the task into multiple adaptive stages provides a more effective refinement process. Even with the same computational budget, multi-stage learning achieves better generative performance, suggesting that the learning scheme itself is crucial to success.

Table 5: Comparison of single-stage vs. multi-stage learning under a fixed total of 15K training iterations.

| Iteration | Single-stage (15K) | Stage 1 (5K) | Stage 2 (5K) | Stage 3 (5K) |
|---|---|---|---|---|
| FID | 15.15 | 18.68 | 12.87 | 9.48 |

**Effect of Langevin step.** Each stage in our framework employs Langevin dynamics to draw samples from the EBM prior refined by the previous stage. Our default setting uses $K = 10$ Langevin steps, but we also examine the effect of varying $K$. As shown in Tab.6, performance improves as $K$ increases: reducing to $K = 5$ yields weaker results (FID = 11.15), while increasing to $K = 15$ produces a substantial gain (FID = 7.11). This trend confirms that more accurate sampling enhances prior quality and improves generative performance. Nevertheless, since larger $K$ values also increase computational overhead, we adopt $K = 10$ as a balanced default across experiments.

Table 6: Impact of Langevin steps $K$ and number of stages $M$ on generation quality for CelebA-HQ-256.

| CIFAR-10 | $K = 5$ | $K = 15$ | $K = 10, M = 3$ | $M = 4$ |
|---|---|---|---|---|
| FID | 11.15 | 7.11 | **7.84** | 7.05 |

**Effect of Stage Number.** We also analyze the influence of the number of stages $M$ in the multi-stage learning framework. With $M = 3$, the model achieves strong generative quality (FID = 7.84). Increasing the number of stages to $M = 4$ further reduces the FID to 7.05, indicating that additional stages continue to refine the prior and improve performance. However, the performance gain is relatively modest compared to the increase in training cost and computational complexity. This observation suggests that while deeper staging can yield incremental improvements, three stages provide an effective balance between accuracy and efficiency. Consequently, we adopt $M = 3$ as the default setting for all reported experiments.

## 6 Conclusion

We proposed a novel framework for learning hierarchical energy-based priors through a multi-stage learning scheme. Instead of relying on a single-stage EBM to approximate the highly complex aggregated posterior, our method progressively refines the prior via a sequence of ratio estimators in a uni-scale latent space. Extensive experiments show that our approach improves image generation quality and also demonstrated its potential in hierarchical controllability, out-of-distribution detection, and energy landscape analysis.

## Acknowledgements

This work is supported in part by NSF IIS-2339604. Any opinions, findings, and conclusions or recommendations expressed in this material are those of the author(s) and do not necessarily reflect the views of the National Science Foundation. We also thank Zhisheng Xiao for sharing the codebase, which facilitated part of this work.

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

# A   Additional Result

**Joint v.s. progressive sampling.**

We compare the two strategies quantitatively. As shown in Tab.7, progressive sampling can yield slightly better generation quality, but at a much higher computational cost (in total 30 steps Langevin dynamics), since each stage requires separate Langevin updates. Joint sampling, in contrast, delivers strong performance with far greater efficiency (in total 10 steps Langevin dynamics), making it the default strategy in our framework. This analysis confirms that our multi-stage hierarchical EBM prior supports both flexible stage-wise refinement and efficient joint sampling.

Table 7: FID for image synthesis of different stages.

| | Prgrogressive Sampling | | | Joint Sampling |
|---|---|---|---|---|
| | Stage 1 | Stage 2 | Stage 3 | |
| CIFAR-10 | 13.15 | 9.72 | 7.62 | 7.80 |
| LSUN-Church-64 | 14.82 | 10.06 | 6.12 | 6.65 |
| CelebA-HQ-256 | 11.48 | 8.65 | 7.71 | 7.84 |

**Stage-wise Analysis with Diffusion Scheme.** Both diffusion-based EBMs Cui & Han (2024) and our multi-stage contrastive framework progressively refine the prior distribution by decomposing the modeling task into simpler intermediate steps. Tab.8 compares the generation quality (FID) of intermediate stages.

While both approaches improve with more refinement stages, our method shows clear advantages. Diffusion-based models rely on fixed schedules and Gaussian perturbations, which restrict flexibility and may lead to inefficient sampling. In contrast, our multi-stage framework is self-adaptive, where each stage learns the density ratio between its evolving prior and the target posterior, which ultimately yields better generative quality.

Table 8: Comparison of diffusion-based and multi-stage priors across refinement steps. $t = (2, 1, 0)$ denotes diffusion steps; $s = (1, 2, 3)$ denotes stages in our multi-stage hierarchical EBM prior.

| FID | Gaussian Prior | $t = 2 / s = 1$ | $t = 1 / s = 2$ | $t = 0 / s = 3$ |
|---|---|---|---|---|
| CIFAR-10 | 37.73 | 18.11 / 13.15 | 11.87 / 9.72 | 8.93 / 7.62 |
| LSUN-Church-64 | 38.13 | 20.15 / 14.82 | 13.54 / 10.06 | 7.31 / 6.12 |
| CelebA-HQ-256 | 30.25 | 17.75 / 11.48 | 12.05 / 8.65 | 8.78 / 7.71 |

**Sample Quality Evolution.** To provide additional evidence of training stability, we track the evolution of sample quality measured by FID. Specifically, we report FID at regular checkpoints both *within* each stage and *across* stages. We compute FID every 2,000 iterations, and every 5,000 iterations we transition to the next stage of learning (denoted by $s = 1, 2, 3$). The results on CIFAR-10 and LSUN-Church-64 are summarized in Table 9. As shown, FID decreases as learning progresses, and importantly, there is no degra-

Table 9: Evolution of generation quality FID (↓) over iterations and stages.

| | $s = 1$ | | $s = 2$ | | | $s = 3$ | |
|---|---|---|---|---|---|---|---|
| | 2000 iter | 4000 iter | 6000 iter | 8000 iter | 10000 iter | 12000 iter | best |
| CIFAR-10 | 21.45 | 14.34 | 12.85 | 10.55 | 9.66 | 7.84 | 7.62 (13500 iter) |
| LSUN-Church-64 | 19.61 | 15.87 | 11.22 | 10.10 | 10.03 | 7.10 | 6.12 (14500 iter) |

dation or fluctuation in quality when switching to the next stage. This improvement across iterations and stages indicates that the dynamic loss behavior observed in Fig.7 is accompanied by consistently improving generation quality.

**Impact of Network Structure and Decoder Type.** In our default configuration, the NVAE backbone employs a discretized logistic decoder. To further evaluate the robustness of our approach, we also experiment with the NVAE backbone using a Gaussian decoder and network structures, consistent with the setting in NCP-VAE Aneja et al. (2021). As shown in Tab.10, our multi-stage learning consistently outperforms the single-stage baseline, achieving substantially lower FID scores. These results demonstrate that the benefits of our multi-stage framework are not tied to a specific decoder choice, but generalize across network structures.

Table 10: Effect of decoder type on CIFAR-10.

| CIFAR-10 | Gaussian prior | Stage 1 | Stage 2 |
|---|---|---|---|
| NCP-VAE | 52.45 | 24.08 | - |
| Ours | 52.45 | 20.68 | 14.04 |

We show additional image synthesis in Fig.11, Fig.12, and Fig.13, as well as Langevin transition results in Fig.14.

## B Derivation

### B.1 Joint Learning Objective

Learning hierarchical generative models (Eqn. 9) can be achieved by maximizing the log-likelihood as $\mathcal{L}_p(\beta, \omega_m) = \frac{1}{N} \sum_{i=1}^{N} \log p_{\beta, \omega_m}(\mathbf{X}_i)$, where $p_{\beta, \omega_m}(\mathbf{X}_i) = \int_{\mathbf{Z}} p_{\beta, \omega_m}(\mathbf{X}_i, \mathbf{Z})d\mathbf{Z}$. For learning the generator parameters $\beta$, the gradient is computed as

$$
\begin{aligned}
\frac{\partial}{\partial \beta}\mathcal{L}_p(\beta, \omega_m) &= \mathbb{E}_{p_{\text{data}}(\mathbf{X})p_{\beta, \omega_m}(\mathbf{Z}|\mathbf{X})}\left[\frac{\partial}{\partial \beta}\log p_{\beta, \omega_m}(\mathbf{X}, \mathbf{Z})\right] \\
&= \mathbb{E}_{p_{\text{data}}(\mathbf{X})p_{\beta, \omega_m}(\mathbf{Z}|\mathbf{X})}\left[\frac{\partial}{\partial \beta}\log p_{\beta_0}(\mathbf{X}|\mathbf{Z})\right] + \mathbb{E}_{p_{\text{data}}(\mathbf{X})p_{\beta, \omega_m}(\mathbf{Z}|\mathbf{X})}\left[\frac{\partial}{\partial \beta}\log p_{\omega_m, \beta_{>0}}(\mathbf{Z})\right] \\
&= \mathbb{E}_{p_{\text{data}}(\mathbf{X})p_{\beta, \omega_m}(\mathbf{Z}|\mathbf{X})}\left[\frac{\partial}{\partial \beta}\log p_{\beta_0}(\mathbf{X}|\mathbf{Z})\right] + \mathbb{E}_{p_{\text{data}}(\mathbf{X})p_{\beta, \omega_m}(\mathbf{Z}|\mathbf{X})}\left[\frac{\partial}{\partial \beta}\log p_{\beta_{>0}}(\mathbf{Z})\right] + C
\end{aligned}
\tag{15}
$$

where $C$ is constant with respect to $\beta$.

For learning the EBM prior $\omega_m$, we adopt contrastive estimation Gutmann & Hyvärinen (2010), i.e., minimizing $-\mathbb{E}_{q(\mathbf{Z})}[\log D_{\omega_m}(\mathbf{Z})] - \mathbb{E}_{p(\mathbf{Z})}[\log(1 - D_{\omega_m}(\mathbf{Z}))]$, which corresponds to learning a density-ratio estimator $r_{\omega_m}(\mathbf{Z}) = \frac{q(\mathbf{Z})}{p(\mathbf{Z})} \approx \frac{D_{\omega_m}(\mathbf{Z})}{1 - D_{\omega_m}(\mathbf{Z})}$. The reparameterized objective yields the learning gradient as

$$
\begin{aligned}
&\frac{\partial}{\partial \omega_m}\left(\mathbb{E}_{q(\mathbf{Z})}[\log D_{\omega_m}(\mathbf{Z})] + \mathbb{E}_{p(\mathbf{Z})}[\log(1 - D_{\omega_m}(\mathbf{Z}))]\right) \\
&= \mathbb{E}_{q(\mathbf{Z})}\left[\frac{\partial}{\partial \omega_m}\log \frac{r_{\omega_m}(\mathbf{Z})}{1 + r_{\omega_m}(\mathbf{Z})}\right] + \mathbb{E}_{p(\mathbf{Z})}\left[\frac{\partial}{\partial \omega_m}\log \frac{1}{1 + r_{\omega_m}(\mathbf{Z})}\right]
\end{aligned}
\tag{16}
$$

Here, we consider the target distribution $q(\mathbf{Z})$ as the generator posterior from the prevous stage, i.e., $p_{\text{data}}(\mathbf{X})p_{\omega_{m-1}, \beta}(\mathbf{Z}|\mathbf{X})$, and $p(\mathbf{Z})$ is the hierarchical EBM prior from previous stage as well, i.e., $p(\mathbf{Z}) = p_{\omega_{m-1}, \beta}(\mathbf{Z})$. These derivations lead to our joint-training objectives in Eqn. 10 and Eqn. 11.

### B.2 Two-phase Learning Objective

In the first-phase learning, we adopt a variational scheme for the hierarchical latent variables by introducing a hierarchical inference model $q_\phi(\mathbf{Z}|\mathbf{X})$ and maximizing the corresponding ELBO (Eqn. 12). This ELBO is equivalent to minimizing the KL divergence between two joint distributions, the data inference joint $p_{\text{data}}(\mathbf{X})q_\phi(\mathbf{Z}|\mathbf{X})$ and the hierarchical generator joint $p_\beta(\mathbf{X}, \mathbf{Z})$, as

$$
D_{\text{KL}}(p_{\text{data}}(\mathbf{X})q_\phi(\mathbf{Z}|\mathbf{X})|p_\beta(\mathbf{X}, \mathbf{Z})) = D_{\text{KL}}(p_{\text{data}}(\mathbf{X})|p_\beta(\mathbf{X})) + \mathbb{E}_{p_{\text{data}}(\mathbf{X})}[D_{\text{KL}}(q_\phi(\mathbf{Z}|\mathbf{X})|p_\beta(\mathbf{Z}|\mathbf{X}))]
\tag{17}
$$

In this decomposition, the first term encourages the generator to match the data marginal over $\mathbf{X}$, while the second term encourages the variational posterior $q_\phi(\mathbf{Z}|\mathbf{X})$ to approximate the true generator posterior $p_\beta(\mathbf{Z}|\mathbf{X})$. As such, we can efficiently draw approximate posterior samples.

Specifically, in the second-phase learning, we treat the posterior samples obtained from the first phase as defining the target distribution in the latent space. Concretely, we draw $\mathbf{Z} \sim q_\phi(\mathbf{Z}|\mathbf{X})$ with $\mathbf{X} \sim p_{\text{data}}(\mathbf{X})$ from the learned first-phase model. If the first-phase model is well-learned, the variational posterior is close

to the generator posterior. This empirical assumption is supported by the strong reconstruction quality shown in Tab.3. In this regime, the induced target distribution over latents can be viewed as the aggregated posterior

$$q(\mathbf{Z}) = \int q_\phi(\mathbf{Z}|\mathbf{X})p_{\text{data}}(\mathbf{X})d\mathbf{X} \approx \int p_\beta(\mathbf{Z}|\mathbf{X})p_{\text{data}}(\mathbf{X})d\mathbf{X} \tag{18}$$

However, this aggregated posterior is typically complex and exhibits a substantial mismatch with the simple Gaussian prior, a phenomenon often referred to as the prior-hole problem Aneja et al. (2021); Cui & Han (2024). The goal of the second phase is therefore to learn a hierarchical EBM prior $p_{\omega,\beta_{>0}}(\mathbf{Z})$ that matches the aggregated posterior and bridge this gap. Maximizing the log-likelihood under the EBM reduces to minimizing the KL divergence:

$$\arg\max_\omega \mathbb{E}_{q(\mathbf{Z})}\left[\log p_{\omega,\beta_{>0}}(\mathbf{Z})\right] = \arg\max_\omega -\mathbb{E}_{q(\mathbf{Z})}\left[\log \frac{q(\mathbf{Z})}{p_{\omega,\beta_{>0}}(\mathbf{Z})}\right] + C = \arg\min_\omega D_{\text{KL}}(q(\mathbf{Z})||p_{\omega,\beta_{>0}}(\mathbf{Z})) \tag{19}$$

where $C$ is the entropy term of $\mathbb{E}_{q(\mathbf{Z})}\left[\log q(\mathbf{Z})\right]$ being constant during this phase. Thus, second-phase learning amounts to minimizing this KL divergence (ours Eqn. 3), aligning the hierarchical EBM prior with the aggregated posterior distribution.

### B.3 Uni-scale Transformation

For conditional Gaussian distribution $p_{\beta_i}(\mathbf{z}_i|\mathbf{z}_{i+1})$, an invertible transformation function $\mathcal{T}_{\beta_{>0}}$ can be defined Xiao et al. (2021). Take 2-layer latent variables as an example,

$$\begin{aligned}
\mathbf{z}_2 &= \mathcal{T}_{\beta_{>0}}^{\mathbf{z}_2}(\mathbf{w}_2) = \mathbf{w}_2 \text{ and} \\
\mathbf{z}_1 &= \mathcal{T}_{\beta_{>0}}^{\mathbf{z}_1}(\mathbf{w}_1, \mathbf{w}_2) = \mu_{\beta_1}(\mathbf{z}_2) + \sigma_{\beta_1}(\mathbf{z}_2) \cdot \mathbf{w}_1
\end{aligned} \tag{20}$$

where $\mathbf{w}_1$ and $\mathbf{w}_2$ are distributed as independent Gaussian noise, i.e., $(\mathbf{w}_1, \mathbf{w}_2) \sim p_0(\mathbf{w}_1, \mathbf{w}_2)$ and $p_0(\mathbf{w}_1, \mathbf{w}_2) = p_0(\mathbf{w}_1)p_0(\mathbf{w}_2)$ with each $p_0(\mathbf{w}_i) \sim \mathcal{N}(0, I_d)$. By the change-of-variable rule, we have

$$\begin{aligned}
p_{\beta_{>0}}(\mathbf{z}_1, \mathbf{z}_2) &= p_0(\mathbf{w}_1, \mathbf{w}_2)|\det(J_{\mathcal{T}_{\beta_{>0}}^{-1}})| \text{ and} \\
p_0(\mathbf{w}_1, \mathbf{w}_2) &= p_{\beta_{>0}}(\mathbf{z}_1, \mathbf{z}_2)|\det(J_{\mathcal{T}_{\beta_{>0}}})|
\end{aligned} \tag{21}$$

where $J_{\mathcal{T}_{\beta_{>0}}}$ is the Jacobian of $\mathcal{T}_{\beta_{>0}}$.

For joint EBM prior on $\mathbf{Z}$-space $p_{\omega,\beta_{>0}}(\mathbf{Z}) = r_\omega(\mathbf{Z}))p_{\beta_{>0}}(\mathbf{Z})$, we can apply the change-of-variable rule and Eqn. 21 as

$$\begin{aligned}
p_{\omega,\beta_{>0}}(\mathbf{W}) &= p_{\omega,\beta_{>0}}(\mathbf{Z})|\det(J_{\mathcal{T}_{\beta_{>0}}})| \\
&= r_\omega(\mathcal{T}_{\beta_{>0}}(\mathbf{W}))p_{\beta_{>0}}(\mathbf{Z})|\det(J_{\mathcal{T}_{\beta_{>0}}})| \\
&= r_\omega(\mathcal{T}_{\beta_{>0}}(\mathbf{W}))p_0(\mathbf{W})
\end{aligned} \tag{22}$$

With such single-stage EBM prior on $\mathbf{W}$-space, we construct our multi-stage EBM prior.

**MCMC Posterior Sampling.** To obtain samples from the generator posterior (i.e., $p_{\omega,\beta}(\mathbf{Z}|\mathbf{X}) \propto p_{\beta_0}(\mathbf{X}, \mathbf{Z})$), the target distribution in the uni-scale $\mathbf{W}$-space is derived as

$$p_{\omega,\beta}(\mathbf{X}, \mathbf{W}) = p_{\beta_0}(\mathbf{X}|\mathcal{T}_{\beta_{>0}}(\mathbf{W}))p_{\omega,\beta_{>0}}(\mathbf{Z})|\det(J_{\mathcal{T}_{\beta_{>0}}})| = p_{\beta_0}(\mathbf{X}|\mathcal{T}_{\beta_{>0}}(\mathbf{W}))p_{\omega,\beta}(\mathbf{W}) \tag{23}$$

**Comparison with Z-space.** In the original $\mathbf{Z}$-space, each $\mathbf{z}_i$ can reside in different scale. In particular, for layers $i \neq j$, the conditional priors $p_{\beta_i}(\mathbf{z}_i|\mathbf{z}_{i+1})$ and $p_{\beta_j}(\mathbf{z}_j|\mathbf{z}_{j+1})$ may have distinct parameterization $\{\mu_{\beta_i}, \sigma_{\beta_i}\}$ and $\{\mu_{\beta_j}, \sigma_{\beta_j}\}$. This induces variations in the magnitude of different scales of $\mathbf{z}_i$ and $\mathbf{z}_j$, resulting in a multi-scale latent space. Sampling in such a space can be challenging: a single global step size is often suboptimal, and one may need carefully designed, scale-adapted updates to ensure stable and efficient exploration. In contrast, our $\mathbf{W}$-space is uni-scale by construction: all $\mathbf{w}$ variables are drawn from a fixed, standard Normal distribution. This enforces a consistent scale across layers and dimensions, greatly simplifying the design of sampling algorithms, reducing the need for ad-hoc scale corrections, and leading to more effective sampling behavior in practice.

### B.4 Coupling with Symbol Vector

Inspired by prior advance Salimans et al. (2016); Pang et al. (2020b), we extend our hierarchical prior with an additional latent–label coupling module to enable controllable generation. After training the hierarchical generator in a fully unsupervised manner in the first phase, we fix the learned generator and learn our hierarchical EBM prior with label vectors $y$. In this way, our model supports controllable generation without re-training the backbone generator.

Specifically, our joint density can be factorized as $p_{\omega_m,\beta_{\geq 0}}(\mathbf{W}, y) = p_{\omega_m,\beta_{\geq 0}}(y|\mathbf{W})p_{\omega_m,\beta_{\geq 0}}(\mathbf{W})$, where $p_{\omega_m,\beta_{\geq 0}}(y|\mathbf{W})$ is a $K$-way classifier (i.e., softmax classifier), and $p_{\omega_m,\beta_{\geq 0}}(\mathbf{W})$ is our hierarchical EBM prior. Such a ratio estimator $r_\omega(\mathcal{T}_{\beta_{>0}}(\mathbf{W}), y)$ can be interpreted as a classifier over $(K+1)$ classes. In practice, we define our ratio estimator as $r_\omega(\mathcal{T}_{\beta_{>0}}(\mathbf{W}), y)$, which produces $K+1$ outputs. The first $K$ outputs correspond to classes $y = \{1 \dots K\}$, and the $(K+1)$-th output plays the role of distinguishing samples from the base distribution (EBM prior of previous stage) versus the target distribution (aggregated posterior).

## C  Discussion of Latent Space Modelling and Learning

Latent variable generative model specifies a joint distribution with a low-dimensional latent space as

$$p_\beta(\mathbf{X}, \mathbf{z}) = p_\beta(\mathbf{X}|\mathbf{z})p_0(\mathbf{z}) \tag{24}$$

where $p_0(\mathbf{z})$ is the prior model usually assumed to be standard Gaussian, and $p_\beta(\mathbf{x}|\mathbf{z})$ is the generation model that maps from low-dimensional latent space to high-dimensional data space. However, such a Gaussian prior model can be non-informative and inexpressive, making the modelling capacity limited and hurting the generative performance.

**Hierarchical Prior.** To tackle this issue, prior arts explore learning a hierarchical prior model by factorizing multiple layers of latent variables (Eqn. 1), such as

$$p_\beta(\mathbf{X}, \mathbf{Z}) = p_{\beta_0}(\mathbf{X}|\mathbf{Z})p_{\beta_{>0}}(\mathbf{Z}) \quad \text{where} \quad p_{\beta_{>0}}(\mathbf{Z}) = \prod_{i=1}^{L-1} p_{\beta_i}(\mathbf{z}_i|\mathbf{z}_{i+1})p_0(\mathbf{z}_L) \tag{25}$$

Such hierarchical generative models or relevant forms have shown strong ability to model complex data distributions and, importantly, to learn multi-level latent representations that reflect semantic hierarchies in the data Sønderby et al. (2016b); Nijkamp et al. (2020); Vahdat & Kautz (2020); Child (2021). However, most of these models still assume each conditional prior to be Gaussian, which primarily captures dependencies across layers, while ignoring the intra-layer relationship, resulting in the hierarchical latent space being under-modelled.

**Energy-based Prior.** Alternatively, the latent variable generative model with EBM prior (LEBM) Pang et al. (2020a) specified a joint distribution

$$p_{\beta,\omega}(\mathbf{X}, \mathbf{z}) = p_\beta(\mathbf{X}|\mathbf{z})p_\omega(\mathbf{z}) \tag{26}$$

$p_\omega(\mathbf{z}) = \frac{1}{Z_\omega} \exp f_\omega(\mathbf{z})p_0(\mathbf{z})$ becomes the EBM prior, which can be more expressive than a Gaussian prior. However, learning EBM prior by MLE requires MCMC sampling, such as Langevin dynamics Neal et al. (2011), as an inner-loop during the training. This motivates follow-up works of different modelling schemes for EBMs (e.g., score-matching Guo et al. (2023)) and learning schemes (e.g., energy discrepancy Schröder et al. (2023), NCE learning Rhodes et al. (2020); Xiao & Han (2022)).

**Hierarchical EBM Prior.** More recently, advances also explore hierarchical EBM prior Cui et al. (2023a); Aneja et al. (2021); Cui et al. (2023b) which intend to leverage the strengths of both the hierarchical generative model and EBM prior. In this setting, the joint distribution is specified as

$$p_{\beta,\omega}(\mathbf{X}, \mathbf{z}) = p_\beta(\mathbf{X}|\mathbf{Z})p_{\omega,\beta_{>0}}(\mathbf{Z}) \tag{27}$$

where $p_{\omega,\beta_{>0}}(\mathbf{Z})$ becomes a hierarchical prior modulated by EBM. However, the aggregated posterior is typically highly multi-modal and strongly structured, and learning a single-stage EBM to approximate the aggregated posterior is difficult, leading to a suboptimal learned hierarchical EBM prior.

**Multi-stage learning.** To address the limitations of single-stage learning, multi-stage EBM frameworks have been proposed. One branch of this line of work is diffusion-based EBM priors Gao et al. (2021); Guo et al. (2023); Cui & Han (2024); Schröder et al. (2023). They model a sequence of gradually perturbed distributions (from simple noise to the target distribution) and learn score functions along a pre-defined noise schedule. These approaches are powerful but typically rely on carefully designed schedules and can be sensitive to hyperparameters such as the number and spacing of noise levels. Another branch is multi-stage NCE, which offers a self-adaptive scheme without a fixed schedule. Specifically, Rhodes et al. (2020) uses a fixed number of stages in data space, but so far shows convincing results only on relatively simple datasets like MNIST. Xiao & Han (2022) introduces adaptive multi-stage schemes in flat latent spaces, but still does not realize the hierarchical structure in the latent representation. Our work builds on this line of research by introducing a multi-stage hierarchical EBM prior.

## D   Broader Impact Statement

This paper proposes a generative probabilistic framework aimed at advancing hierarchical representation learning and controllable generation, and thus shares similar potential risks as other powerful generative models. In particular, fine-grained control over semantic attributes could, in principle, be misused to create manipulated or misleading visual content. At the same time, our work on OOD detection may help mitigate some risks by enabling more reliable detection of anomalous or out-of-distribution inputs in downstream systems.

## E   Implementation and Algorithm

### E.1   Implementation

**Training detail.** For joint learning, we benchmark our method on CIFAR-10, SVHN, and CelebA-64, for which we scale training images to $[-1, 1]$ and use only 40,000 examples of CelebA-64 following Xiao & Han (2022). For Two-phase training, we train our model on CIFAR-10, LSUN-Church-64, and CelebA-HQ-256 using the same setting as NVAE Vahdat & Kautz (2020). We compute FID scores using 30,000 generated images for CelebA-HQ-256 and 50,000 for other datasets.

We use one A100 Nvidia GPU (training time on CIFAR-10 for 0.15 seconds/iteration) for joint learning and two A100 Nvidia GPUs (training time on CIFAR-10 for 8.45 seconds/iteration) for Two-phase training. Please see further implementation details at `https://github.com/jcui1224/multi-stage-nce-ebm`.

**Network Structure.** For joint learning, we adopt the network structures from Xiao & Han (2022), which contain generator and energy function networks shown in Tab.12. For Two-phase learning scheme, we utilize the NVAE[2] backbone model, and our energy function is shown in Tab.11.

---

[2]https://github.com/NVlabs/NVAE

## E.2 Training Algorithm

---

**Algorithm 1** Joint Learning Scheme

---

**Require:**

Training images $\mathbf{x}$; Training iteration $i$; Each stage iteration $I$; Current stage number $m$; Total stages $M$; Joint EBM prior $\omega_m$; Hierarchical generator model $\beta$; Posterior Buffer $B_m$ and $B_{m-1}$; Langevin steps $k$;.

1: Let $m \leftarrow 0$, initialize $\omega_m$ and $\beta_m$.
2: **repeat**
3:     **repeat**
4:         **Posterior Sample:** obtain $\mathbf{W}_{\text{true}} \sim p_{\omega_m,\beta}(\mathbf{Z}|\mathbf{X})$ using Eqn.14 with $k$.
5:         **Save Buffer:** Save posterior sample $\mathbf{W}_{\text{true}}$ to Posterior Buffer $B_m$.
6:         **Prior sample:** Draw $\mathbf{W}_{\text{fake}}$ from $B_{m-1}$ if $m > 0$ else $\mathbf{W}_{\text{fake}} \sim N(0, I)$
7:         **Learn $\beta_m$:** Update $\beta_m$ with $\mathbf{W}_{\text{true}}$ using $\mathbf{Z}_{\text{true}} = \mathcal{T}_{\beta>0}(\mathbf{W}_{\text{true}})$ and Eqn. 10
8:         **Learn $\omega_m$:** Update $\omega_m$ with $\mathbf{Z}_{\text{true}}$ (left term in Eqn. 11) and $\mathbf{Z}_{\text{fake}} = \mathcal{T}_{\beta>0}(\mathbf{W}_{\text{fake}})$ (right term in Eqn. 11).
9:     **until** $i = I$
10:     Replace $B_{m-1}$ by $B_m$ and initiate new $B_m$
11: **until** $m = M$

---

**Algorithm 2** Two-phase Scheme

---

**Require:**

Training images $\mathbf{x}$; First-phase training iteration $i_1$; First-phase total iteration $I_1$; Hierarchical model $\beta, \phi$; Second-phase training iteration $i_2$; Each stage iteration $I_2$; Current stage number $m$; Total stages $M$; Joint EBM prior $\omega_m$; Langevin steps $k$;. Prior Buffer B;

1: **(1) First-phase learning:** initialize $\beta, \phi$.
2: **repeat**
3:     **Variational learning for $\beta, \phi$:** update $\beta, \phi$ via Eqn. 12
4: **until** $i = I_1$
5: **(2) Second-phase learning:** Fix $\beta, \phi$ and initialize $m = 0$, $\omega_m$.
6: **repeat**
7:     if $m > 0$, prepare Buffer $B_{m-1}$ by EBM prior sampling via Eqn.14 with $k$ else continue.
8:     **repeat**
9:         **Posterior Sample:** obtain $\mathbf{Z}_{\text{true}} \sim q_\phi(\mathbf{Z})$ and $\mathbf{W}_{\text{true}} = \mathcal{T}_{\beta>0}^{-1}(\mathbf{Z}_{\text{true}})$
10:         **Prior sample:** Draw $\mathbf{W}_{\text{fake}}$ from $B_{m-1}$ if $m > 0$ else $\mathbf{W}_{\text{fake}} \sim N(0, I)$
11:         **Learn $\omega_m$:** Update $\omega_m$ with $\mathbf{Z}_{\text{true}} = \mathcal{T}_{\beta>0}(\mathbf{W}_{\text{true}})$ (left term in Eqn. 13) and $\mathbf{Z}_{\text{fake}} = \mathcal{T}_{\beta>0}(\mathbf{W}_{\text{fake}})$ (right term in Eqn. 13).
12:     **until** $i_2 = I_2$
13: **until** $m = M$

---

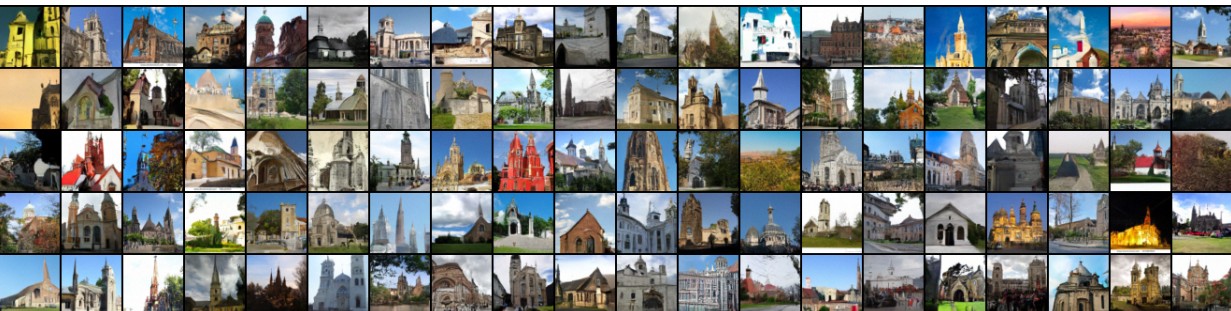

Figure 11: Additional synthesis on LSUN-Church-64.

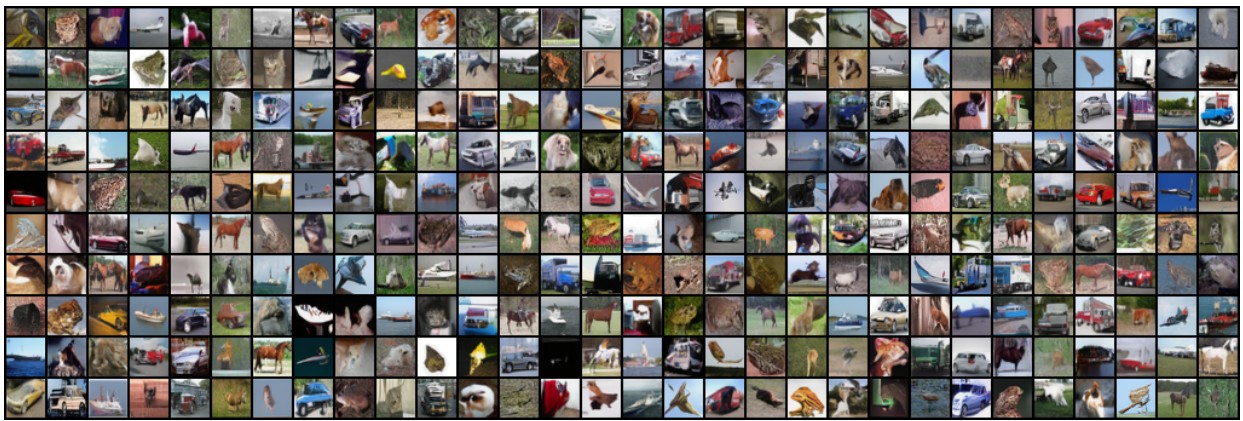

Figure 12: Additional synthesis on CIFAR-10.

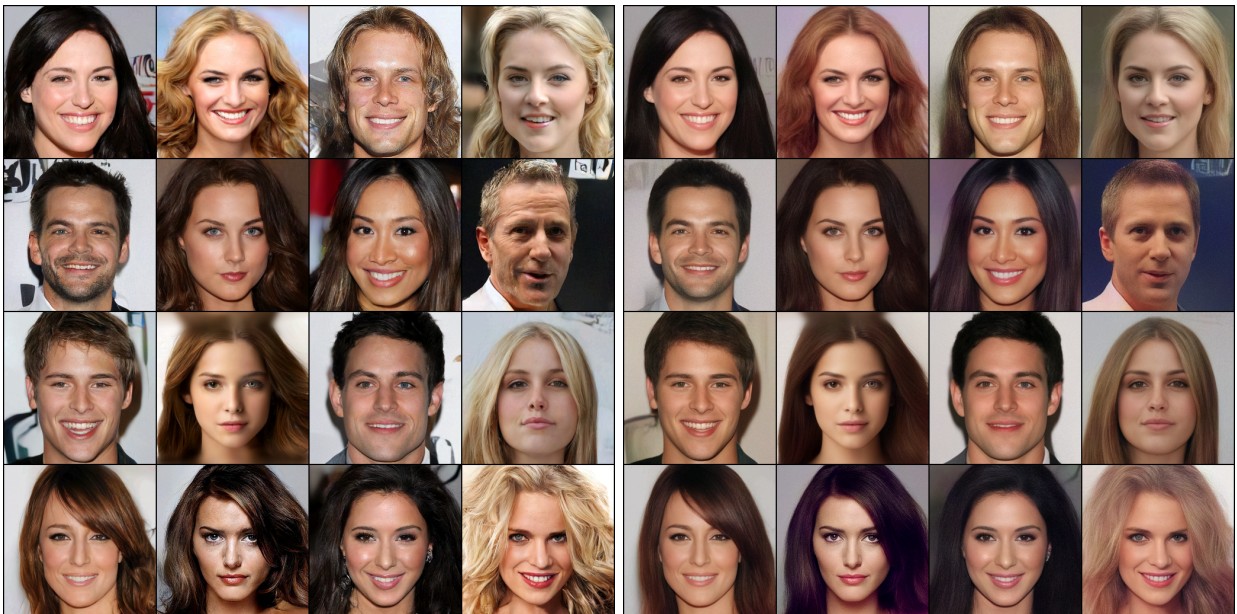

Figure 13: Additional synthesis on CelebA-HQ-256 with temp= 1.0 (left) and temp= 0.7 (right).

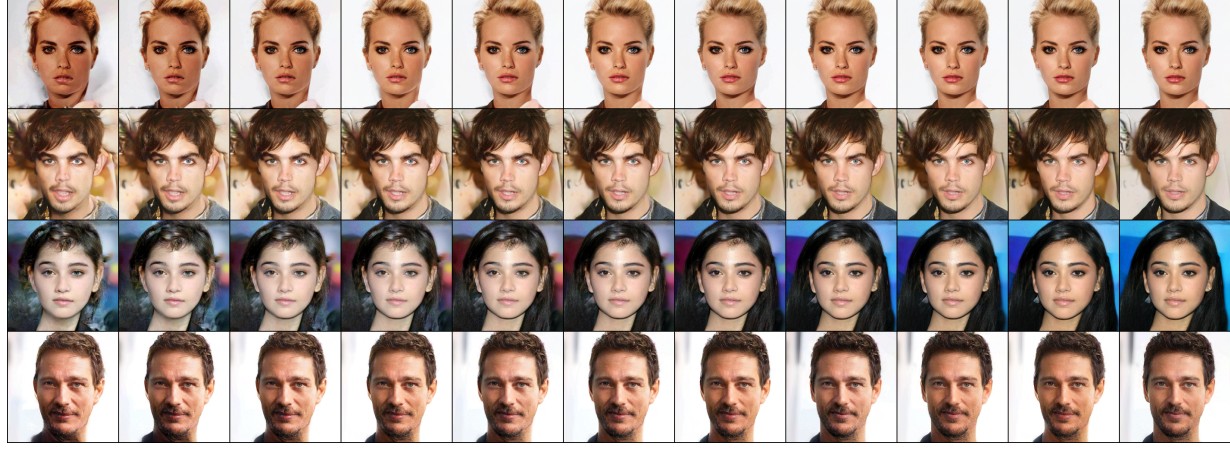

Figure 14: Additional Langevin transition on LSUN-Church-64, CIFAR-10, and CelebA-HQ-256.

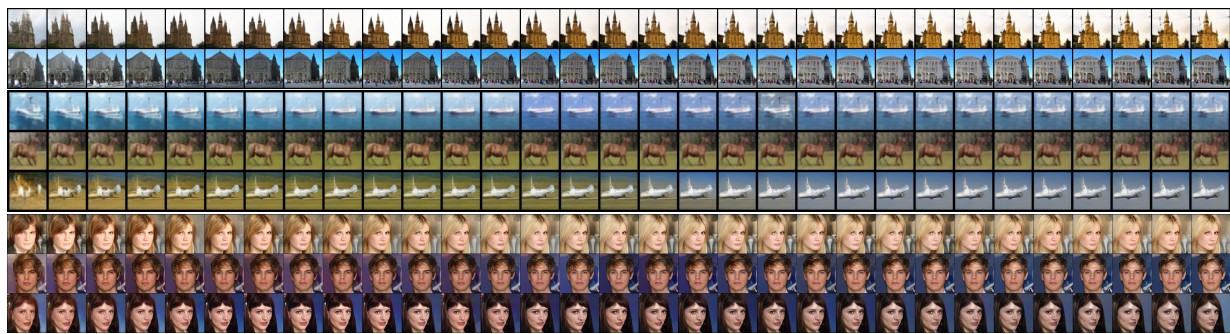

Figure 15: Additional progressive Langevin transition on LSUN-Church-64, CIFAR-10, and CelebA-HQ-256.

Table 11: Two-phase learning energy function on each $\mathbf{z}_i$.

| Conv Residule Block (in-ch, out-ch, downsample, preActivation) | |
|---|---|
| Input: SiLU($h_0$) if preActivation else $h_0$ | |
| $h_1(h_0) = $ 4x4 conv (out-ch), s=2, SiLU | if downsample |
| $h_1(h_0) = $ 3x3 conv (out-ch), s=1, SiLU | if not downsample |
| $h_1(h_1) = $ 3x3 conv (out-ch) | if in-ch≠out-ch or downsample |
| $h_{\text{short}}(h_0) = $ 4x4 conv (out-ch), s=2 | if downsample |
| $h_{\text{short}}(h_0) = $ 3x3 conv (out-ch), s=1 | if in-ch≠out-ch |
| $h_{\text{short}}(h_0) = h_0$ | else |
| return $h_1 + h_{\text{short}}$ | |

| Conv Residule (in-ch, out-ch, res-layer) |
|---|
| Input: $h_0$ |
| Conv Residule Block (in-ch, out-ch, downsample=True, preActivation=True) |
| #res-layer x Conv Residule Block (in-ch, out-ch, downsample=False, preActivation=True) |

| Linear Residule Block (in-ch, out-ch, preActivation) | |
|---|---|
| Input: SiLU($h_0$) if preActivation else $h_0$ | |
| $h_1(h_0) = $ Linear (out-ch), SiLU | - |
| $h_1(h_1) = $ Linear (out-ch) | if in-ch≠out-ch |
| $h_{\text{short}}(h_0) = $ Linear (out-ch) | if in-ch≠out-ch |
| $h_{\text{short}}(h_0) = h_0$ | else |
| return $h_1 + h_{\text{short}}$ | |

| Energy function (nef=128, ndf=256, res-layer=8) | |
|---|---|
| Layer | In-Out Size |
| Input: $\mathbf{w}_i$ | h x w x ch |
| 3x3 conv (nef), s=1 | h x w x nef |
| # of Conv Residule (nef, nef, res-layer) | 4 x 4 x 64 |
| flatten | 4 * 4 * 64 |
| # res-layer of Linear Residule Block (ndf, ndf, preActivation=True) | ndf |
| SiLU, Linear (1) | 1 |

Table 12: 2-layer generator network on CIFAR-10.

| $p_{\beta_0}(\mathbf{x}|\mathbf{z}_1)$, ngf= 64 | | |
|---|---|---|
| Layers | In-Out Size | Stride |
| Input: $\mathbf{z}_1$ | 1 x 1 x 100 | - |
| 4x4 convT (ngf x 8), LReLU | 4 x 4 x (ngf x 8) | 1 |
| 4x4 convT (ngf x 4), LReLU | 8 x 8 x (ngf x 4) | 2 |
| 4x4 convT (ngf x 2), LReLU | 16 x 16 x (ngf x 2) | 2 |
| 4x4 convT (3), Tanh | 32 x 32 x 3 | 2 |

| $p_{\beta_1}(\mathbf{z}_1|\mathbf{z}_2)$, ngf= 256 | |
|---|---|
| Layers | In-Out Size |
| Input: $\mathbf{z}_2$ | 100 |
| Linear (ngf), LReLU | ngf |
| Linear (ngf), LReLU | ngf |
| Linear (ngf), LReLU | ngf |
| Linear (ngf), LReLU | ngf |
| Linear (100), $\mu$ of $\mathbf{w}_1$ | 100 |
| Linear (100), $\log \sigma$ of $\mathbf{w}_1$ | 100 |

| energy function for each $\mathbf{w}_i$, ndf= 200 | |
|---|---|
| Layers | In-Out Size |
| Input: $\mathbf{z}_1$ | 100 |
| Linear (ngf), LReLU | ndf |
| Linear (ngf), LReLU | ndf |
| Linear (ngf), LReLU | ndf |
| Linear (ngf), LReLU | ndf |
| Linear (1) | 1 |

