# OpenReview forum: "Learning Adaptive Multi-Stage Energy-based Prior for Hierarchical Generative Model"
_TMLR — Accepted by TMLR_

### Review · Reviewer_3mes · 2025-11-12

**Summary Of Contributions:**

This paper introduces a framework for learning multi-stage hierarchical Energy-Based Model (EBM) priors to resolve the prior-hole problem in hierarchical generative models (HGMs). Instead of relying on a single, less expressive Gaussian prior, the method uses a sequence of density-ratio estimators to progressively refine the EBM prior and match the complex, multi-modal aggregated posterior. The approach achieves superior generative performance and yields richer hierarchical representations, enabling both generative modelling and out-of-distribution detection.

Although most of the techniques employed in this work are based on prior studies. For example:

- the multi-scope NCE approach from [1],
- the “uni-scale” sampling trick for latent EBMs from [2]
- the multi-stage latent EBM training strategy from [3].

The integration of these methods to scale EBMs for image modeling and achieve performance competitive with diffusion models makes this paper promising.

[1] Rhodes, Benjamin, Kai Xu, and Michael U. Gutmann. "Telescoping density-ratio estimation." *Advances in neural information processing systems* 33 (2020): 4905-4916.

[2] Xiao, Zhisheng, et al. "Vaebm: A symbiosis between variational autoencoders and energy-based models." arXiv preprint arXiv:2010.00654 (2020).

[3] Xiao, Zhisheng, and Tian Han. "Adaptive multi-stage density ratio estimation for learning latent space energy-based model." Advances in Neural Information Processing Systems 35 (2022): 21590-21601.

**Audience:**

Yes

**Audience Explanation:**

Yes. Researchers working on EBMs will likely appreciate the proposed scalable multi-stage learning paradigm. It is encouraging to see that EBMs can be effectively scaled for image modeling. Although the methodology largely integrates existing techniques, the resulting strong performance is still impressive and valuable to the community.

**Broader Impact Concerns:**

There are no broader impact concerns.

**Claims And Evidence:**

Yes

**Claims Explanation:**

The claims are supported by extensive and well-structured experimental evidence.

**Requested Changes:**

- The justification for the objective function in Equation 4 could be strengthened. It wasn't immediately clear what the theoretical connection is between this equation and the principle of maximum likelihood. A detailed discussion in the appendix would be a valuable addition.
- The term "uni-scale" introduced in Section 2 requires clarification. While I suspect this terminology may be used because the variable $W$ resides in a Gaussian noise space, the paper would benefit from a more detailed and explicit explanation of what "uni-scale" means and why this property is relevant.
- I would suggest including the derivations for Equations 11 and 12 in the appendix to make the paper more self-contained. The paper [4] (see Appendix C.1) provides a very detailed discussion on various methods for training LEBMs (e.g., CD, NCE, SM). Incorporating a similar, brief discussion of these training paradigms would be helpful for the reader.
- In Equation 14, the posterior p(Z|X) is replaced with a variational distribution, but the EBM prior p(Z) still requires MCMC sampling. It would be helpful to explicitly clarify this point in the paper.
- Following up on this observation, it is unclear why only p(Z|X) is replaced with a variational distribution, while p(Z) is still modeled via MCMC. Have the authors considered or conducted an ablation study comparing the following configurations:
    - Both p(Z|X) and p(Z) modeled with variational approximations
    - Both p(Z|X) and p(Z) modeled with MCMC
    - Variational for p(Z|X) but MCMC for p(Z) (the current approach)
    - Variational for p(Z) but MCMC for p(Z|X)
- It would also improve clarity if the paper included a concise algorithm summarising the overall training and inference procedures.

[4] Schröder, Tobias, et al. "Energy discrepancies: a score-independent loss for energy-based models." *Advances in Neural Information Processing Systems* 36 (2023): 45300-45338.

---

> ### Author Response · Authors · 2025-12-26
>
> We thank the reviewer for the constructive comments and helpful references. We have revised the manuscript accordingly and discussed with the reference and compared it in the **Table. 1**.
>
> In particular,
>
> - Detailed derivations for Eqn. 4, 11 and 12 are now provided in **Sec. B.1** and **Sec. B.2**.
>
> - Explanation of the term “uni-scale” is included in **Sec. B.3**.
>
> - A broad discussion with related works is provided in **Sec. C**.
>
> - Algorithms for our proposed method are added in **Sec. E.2**.
>
>
>
> Our paper studies a self-adaptive multi-stage learning scheme for hierarchical EBM priors. We employ two complementary regimes depending on the hierarchical structure used. Below, we further clarify our design choices regarding when MCMC sampling or variational approximation is used.
>
> **(i) Shallow hierarchy:** When the hierarchy is shallow (e.g., 2 layers), the generator and the EBM prior can be trained jointly with MCMC posterior sampling. In this setting, the computational cost of backpropagating through the generator is manageable, and the MCMC-based posterior sampling can be more accurate and can provide meaningful learning signals to the latent distribution. To further improve efficiency, the posterior samples produced at the previous stage are **reused** as approximations to EBM prior samples in the next stage, which facilitates an efficient learning scheme [3].
>
> **(ii) Deep hierarchy:** When the hierarchy becomes deep (e.g., 30 layers), MCMC posterior sampling becomes computationally expensive due to repeated backpropagation through the deep generative model $p_{\beta_0}(X|Z)$, which is typically large. Jointly updating both the hierarchical generator and hierarchical EBM prior under such conditions can be unstable and challenging. We take inspiration from prior arts [2] and adopt a two-phase learning strategy in which a hierarchical variational inference model replaces the MCMC posterior sampler. This choice provides tractable learning dynamics in deep hierarchical structures. We emphasize that our hierarchical EBM prior does not incur inner-loop MCMC sampling. MCMC EBM sampling is performed only when transitioning from stage $m$ to stage $m+1$ in order to construct a bucket of prior samples (i.e., samples from base distribution) used for the next stage. These samples remain fixed during the current stage of learning and do not introduce iterative computational overhead. As a result, the total cost associated with prior sampling is minimal relative to the overall training pipeline.
>
> We hope this explanation clarifies our principled choice of combining MCMC and variational inference within the multi-stage learning framework, balancing expressive modeling with computational efficiency. Exploring fully variational alternatives for both $p(Z \mid X)$ and $p(Z)$ is an interesting direction, which we leave for future work.

---

### Review · Reviewer_GZj5 · 2025-12-09

**Summary Of Contributions:**

The paper tackles the "prior-hole" problem in hierarchical generative models by introducing a Multi-Stage Hierarchical Energy-Based Model (EBM) Prior. The proposed framework is mathematically solid, effectively decomposing the difficult density ratio estimation task into a sequence of adaptive, manageable stages. The authors also introduce a technically sound "uni-scale" reparameterization to address the challenges of multi-scale latent spaces. The experimental results are promising, demonstrating that this mathematically grounded approach consistently improves generation quality over single-stage baselines.

**Audience:**

Yes

**Audience Explanation:**

The paper presents a solid theoretical contribution to the intersection of VAEs and EBMs. The rigorous mathematical formulation of the multi-stage density ratio estimation in hierarchical spaces will be of significant interest to researchers focusing on rigorous probabilistic modeling and latent space dynamics.

**Broader Impact Concerns:**

The authors discuss "Fine-grained Controllability," which allows for the manipulation of specific attributes like gender or appearance. While this is a technical contribution, it raises standard concerns regarding the generation of manipulated media. I request that the authors include a brief Broader Impact Statement acknowledging the potential misuse of such controllability features, alongside the positive applications of their OOD detection work.

**Claims And Evidence:**

Yes

**Claims Explanation:**

The experimental section is comprehensive and the results are promising. The comparisons against various baselines (including single-stage EBMs and diffusion models) provide good evidence that the multi-stage refinement strategy is effective. The ablation studies further support the mathematical intuition behind the stage-wise decomposition.

**Requested Changes:**

I recommend the following adjustments to strengthen the submission:

1.  **Evidence of Stability in Joint Training:** The paper emphasizes the stability of the joint learning scheme and points to the smooth loss trends in Figure 8 as evidence. However, in contrastive or adversarial settings, a stable training loss does not strictly guarantee better generation quality (e.g., it could indicate mode collapse). Can the authors provide further evidence of stability beyond the loss curves? For instance, monitoring the variance of FID scores across different training checkpoints or providing visual samples over time would be much more convincing evidence that the "stable" loss correlates with stable generation capability.
2.  **Clarification on Controllability:** In Section 5.2.1, the paper discusses "Hierarchical Controllability" and mentions extending the EBM into a multi-class classifier using "K class labels." However, the text also claims the backbone generator is trained in a "fully unsupervised manner." This creates confusion regarding the supervision setting.
    * How exactly are the concepts of gender or attributes "injected" into the model?
    * Does the EBM training require labeled data while the generator does not?
    * If labels are used, please clarify this distinction to ensure the claims about unsupervised learning are accurate and the method's data requirements are clear.

---

> ### Author Response · Authors · 2025-12-26
>
> We thank the reviewer for their insightful comments. We have updated our manuscript for the broader impact in the **Sec. D**.
>
> **(i) Evidence of stability beyond loss curves:** In the revised manuscript, we provide additional evidence in **Appendix A**. Specifically, we track the evolution of sample quality over training by reporting FID at regular checkpoints both within and across stages (every 2,000 iterations, with stage transitions every 5,000 iterations). The results in **Table 10** show a consistent improvement in FID as training progresses, with no degradation when moving between stages. Taken together with the smooth loss trends in Fig. 8, these results indicate stable learning dynamics and steadily improving generation quality.
>
> **(ii) Clarification on Controllability:** For the controllable generation experiment, our pipeline is explicitly two-step:
>
> - In the **first phase**, the hierarchical generator and latent hierarchy are trained in a fully unsupervised manner, using only images (no attribute labels). At this stage, the model is only feasible for unconditional/random generation.
> - In the **second phase**, we introduce a hierarchical EBM prior that is coupled with label vectors $y$ to enable controllable generation, while keeping the first-phase hierarchical generator fixed. Concretely, we define an additional latent–label coupling module and train this module using attribute labels. Thus, labels are used only to train the hierarchical EBM prior, without retraining or fine-tuning the generator itself.
>
> We have clarified this distinction in **Sec. B.4** of the revised manuscript, explicitly stating that *(i)* the generative model is trained without labels, and *(ii)* labels are used only in the second-phase latent–label coupling module for controllable generation.
>
> We hope this addresses the concerns regarding both stability and the supervision setting for controllability.

---

### Review · Reviewer_xRFw · 2025-12-13

**Summary Of Contributions:**

This paper proposes a novel framework that introduces multi-stage energy-based models (EBMs) priors for hierarchical generative models. The authors claim that this framework can enable more expressive latent modeling than a Gaussian or single-stage EBM prior. To improve the effectiveness and efficiency, the authors use a uni-scale latent space reparameterization and develop two training schemes.

**Audience:**

Yes

**Audience Explanation:**

This paper introduces a multi-stage energy-based prior for hierarchical generative models. It provides valuable insights for subsequent researchers, both in enhancing the expressiveness of latent modeling and in designing more efficient generation pipelines.

**Claims And Evidence:**

Yes

**Claims Explanation:**

This paper provides a detailed explanation of the method they proposed, and presents visual effects and quantitative comparisons in the paper.

**Requested Changes:**

There are some concerns and questions:

1. The author only conducted experiments on small-scale datasets such as CelebA-HQ-256, LSUN-Church-64, and CIFAR10. At present, the generation models need to undergo experiments on ImageNet, such as the papers like MAR, SiD, and DiT. So, could the author provide an experimental result on ImageNet?

2. Considering that this paper was submitted in late 2025, the most recent baseline methods included in the comparisons were published in 2024. This may somewhat limit the assessment of the proposed method’s effectiveness. Could the authors consider adding comparisons with more recent methods?

**I am not an expert in the field of Hierarchical Generative Model. For the novelty of this paper, please refer to the opinions of other reviewers.**

---

> ### Author Response · Authors · 2025-12-26
>
> We thank the reviewer for the constructive comments and helpful suggestions.
>
> We have updated **Tables 1 and 3** in the revised manuscript to include more recent baselines, where our method still demonstrates superior performance.
>
> **Comparison with State-Of-The-Art:** Our primary goal in this work is ***not*** to achieve state-of-the-art image generation quality, but to develop and analyze a new learning framework for hierarchical generative models with hierarchical EBM priors. For this purpose, we focus on widely used benchmarks for hierarchical latent variable models, such as CIFAR-10, LSUN-Church-64, and CelebA-HQ-256. As reflected in our updated comparisons (Table 1 and 3), our method still demonstrates superior performance on these datasets, indicating that they remain challenging and informative benchmarks for evaluating modeling quality in this setting.
>
> We hope this clarifies the concerns about the scope and positioning of our work.

---

### Decision · Action_Editor_Paa6 · 2026-01-20

**Recommendation:** Accept with minor revision

**Additional Comments:**

Reviewers commended the proposed idea as interesting and clearly presented. They suggested a few more experiments to conduct to back up some claims (e.g., stability of joint training, the choice of variational approaches for some components but MCMC for some other components). While the first point has been addressed in revision (appendix A), the latter has not been addressed in revision (but with a short comment in reply). I suggest the authors to address this point, preferably with an experiment, in the camera ready.

Some clarifications on the detailed derivations have been provided in revision (in appendix). I suggest to keep them and mention these results in main text for proper signposting.

**Audience:**

Yes

**Audience Explanation:**

Machine learning researchers interested in generative models, especially VAEs and EBMs, would find this paper interesting to read.

**Claims And Evidence:**

Yes

**Claims Explanation:**

This paper studies the problem of learning priors for hierarchical generative models by proposing a multi-stage hierarchical EBM as a prior. To learn this prior, this paper proposes using multi-stage density ratio estimation to for an NCE-like procedure. Other useful tricks proposed in the paper include reparameterisation of the latent EBM prior, and the multi-phase training strategy for the joint model.

Experiments show the model can generate images with decent qualities, and the analyses show interesting representation learning applications such as hierarchical controllable synthesis and OOD detection. Some ablation studies are also provided on the multi-phase training strategy.